# Ferroptotic stress promotes the accumulation of pro-inflammatory proximal tubular cells in maladaptive renal repair

**Shintaro Ide[1†], Yoshihiko Kobayashi[2†], Kana Ide[1], Sarah A Strausser[1], Koki Abe[1], Savannah Herbek[1], Lori L O'Brien[3], Steven D Crowley[1], Laura Barisoni[1,4], Aleksandra Tata[2], Purushothama Rao Tata[2,5,6], Tomokazu Souma[1,5]***

[1]Division of Nephrology, Department of Medicine, Duke University School of Medicine, Durham, United States; [2]Department of Cell Biology, Duke University School of Medicine, Durham, United States; [3]Department of Cell Biology and Physiology, University of North Carolina at Chapel Hill, Chapel Hill, United States; [4]Department of Pathology, Duke University School of Medicine, Durham, United States; [5]Regeneration Next, Duke University, Durham, United States; [6]Duke Cancer Institute, Duke University School of Medicine, Durham, United States

**Abstract** Overwhelming lipid peroxidation induces ferroptotic stress and ferroptosis, a non-apoptotic form of regulated cell death that has been implicated in maladaptive renal repair in mice and humans. Using single-cell transcriptomic and mouse genetic approaches, we show that proximal tubular (PT) cells develop a molecularly distinct, pro-inflammatory state following injury. While these inflammatory PT cells transiently appear after mild injury and return to their original state without inducing fibrosis, after severe injury they accumulate and contribute to persistent inflammation. This transient inflammatory PT state significantly downregulates glutathione metabolism genes, making the cells vulnerable to ferroptotic stress. Genetic induction of high ferroptotic stress in these cells after mild injury leads to the accumulation of the inflammatory PT cells, enhancing inflammation and fibrosis. Our study broadens the roles of ferroptotic stress from being a trigger of regulated cell death to include the promotion and accumulation of proinflammatory cells that underlie maladaptive repair.

*For correspondence:
tomokazu.souma@duke.edu

†These authors contributed equally to this work

Competing interests: The authors declare that no competing interests exist.

## Introduction

Acute kidney injury (AKI) afflicts 1.2 million hospitalized patients annually in the US; 20–50% of AKI survivors progress to chronic kidney disease (CKD), increasing their risk for dialysis-dependency, cardiovascular events, and mortality (*Chawla et al., 2014*; *Lewington et al., 2013*; *Strausser et al., 2018*). Other than general supportive care, there are no targeted therapies to treat AKI or to prevent AKI to CKD transition. A better understanding of the molecular events underpinning the AKI to CKD transition is needed to develop therapeutic strategies to interrupt this devastating disease process.

Clinical and preclinical studies have identified damage to proximal tubular (PT) epithelial cells after severe AKI as a critical mechanism driving transition to CKD (*Strausser et al., 2018*; *Chawla et al., 2011*; *Liu et al., 2017*; *Cippà et al., 2018*; *Ferenbach and Bonventre, 2015*; *Humphreys, 2018*). PT cells are most severely affected by acute ischemic and toxic injuries due to their high metabolic and energy-intensive transporter activities required to maintain normal homeostasis of body fluids (*Ferenbach and Bonventre, 2015*; *Humphreys, 2018*; *Gewin, 2018*). In the renal

repair process, damaged PT cells adopt heterogeneous molecular states (*Kirita et al., 2020*). They reactivate genes normally active during renal development (*Rudman-Melnick et al., 2020*; *Kang et al., 2016*; *Kumar et al., 2015*), alter their dependency on metabolic fuels (*Legouis et al., 2020*), change their morphology, and proliferate to replenish the areas of denuded epithelium in the proximal tubule (*Ferenbach and Bonventre, 2015*; *Witzgall et al., 1994*). When the initial damage to kidneys is mild, PT cells subsequently return to their original state by redifferentiation, with resolution of inflammation and fibrosis (*Ferenbach and Bonventre, 2015*; *Kirita et al., 2020*; *Rudman-Melnick et al., 2020*; *Legouis et al., 2020*; *Witzgall et al., 1994*; *Berger et al., 2014*; *Kusaba et al., 2014*). However, if damage is more extensive, prolonged, or recurrent, the damaged cells fail to redifferentiate, leading to persistent inflammation, fibrosis, and eventual cell death. The molecular pathways that govern proximal tubular heterogeneity and cell fate during failed renal repair after severe injury are poorly understood. This knowledge gap prevents the development of therapies based on underlying disease mechanisms.

One of the critical pathways involved in AKI pathogenesis and proximal tubular cell death is ferroptosis, a distinct non-apoptotic form of regulated cell death (*Stockwell et al., 2017*; *Dixon et al., 2012*; *Yang et al., 2014*; *Kagan et al., 2017*; *Doll et al., 2017*; *Alim et al., 2019*; *Zhao et al., 2020*; *Linkermann et al., 2014*). An imbalance between the generation of lipid peroxides and their detoxification induces overwhelming accumulation of lipid peroxides (ferroptotic stress), triggering ferroptosis (*Stockwell et al., 2017*; *Alim et al., 2019*). The glutathione/glutathione peroxidase 4 (GPX4) axis is the central defense pathway to prevent ferroptotic stress and ferroptosis (*Stockwell et al., 2017*; *Dixon et al., 2012*; *Yang et al., 2014*; *Friedmann Angeli et al., 2014*). Global genetic deletion of *Gpx4* in mice causes renal tubular epithelial death and acute kidney injury, identifying renal tubular epithelial cells as one of the cell types most vulnerable to ferroptotic stress (*Friedmann Angeli et al., 2014*). Moreover, reduced glutathione and NADPH availability further render ischemia-reperfusion injured kidneys vulnerable to ferroptotic stress (*Strausser et al., 2018*; *Nezu et al., 2017*). Accumulating evidence suggests that pharmacological inhibition of ferroptotic cell death ameliorates AKI severity and excess ferroptotic stress has been linked to failed renal repair in patients, suggesting a new therapeutic target (*Zhao et al., 2020*; *Linkermann et al., 2014*; *Friedmann Angeli et al., 2014*; *Wenzel et al., 2017*). Interestingly, recent evidence suggests that molecular regulators of necroptosis, another form of regulated cell death, contribute to disease pathogenesis by additional pathways independent of their well-documented roles in triggering cell death (*Daniels et al., 2017*; *Moriwaki and Chan, 2016*; *Moriwaki et al., 2014*). However, it is still not clear whether ferroptotic stress has additional roles in the pathogenesis of AKI and its sequelae beyond the induction of ferroptotic cell death and loss of functional tubular cells.

Here, using complementary single-cell transcriptomic and mouse genetic approaches, we identify the role of a molecularly distinct, damage-associated, PT cell state that is dynamically and differentially regulated during successful versus failed repair. Furthermore, we provide mechanistic evidence that ferroptotic stress in PT cells enhances this damage-associated state, in addition to triggering cell death, thereby promoting failed renal repair and the AKI-to-CKD transition.

## Results

### Tubular epithelial cells exhibit heterogeneous molecular states after severe injury

To identify cellular mechanisms that promote maladaptive repair after severe kidney injury, we first developed and optimized mouse models for 'successful' versus 'failed' renal repair after ischemia-reperfusion-induced injury (IRI). This was achieved by extending renal ischemic times from 20 min for successful recovery to 30 min for failed recovery (*Figure 1—figure supplement 1* and *Figure 1—figure supplement 2*). After mild injury, histologic examination showed that inflammation and macrophage accumulation resolved within 21 days (ischemic time 20 min; *Figure 1—figure supplement 1, D and E*). By contrast, after severe injury there was progressive epithelial damage and fibrosis, and the accumulation of F4/80+ macrophages persisted around the damaged epithelial cells for at least 6 months (ischemic time 30 min; *Figure 1—figure supplement 1, D and E*; and *Figure 1—figure supplement 2E*).

We used this failure-to-repair model (unilateral IRI, ischemic time 30 min) to generate a single-cell transcriptome map of failed renal repair (*Figure 1A*). Kidneys were harvested at 6 hr and 1, 7, and 21 days after IRI. High-quality transcriptome data from a total of 18,258 cells from injured kidneys (IRI) and homeostatic uninjured kidneys (Homeo) were obtained (*Figure 1*, B and C). Using a Seurat integration algorithm that normalizes data and removes potential batch effects (*Stuart et al., 2019*; *Hafemeister and Satija, 2019*), we integrated the transcriptome data from each condition and performed unsupervised clustering analysis of the integrated dataset. Uniform manifold approximation and projection (UMAP) resolved 21 separate clusters, representing distinct cell types (*Figure 1B*; *Figure 1—figure supplement 3B* and *Figure 1—figure supplement 4A*). The cellular identity of each cluster was determined based on known cell-type-specific markers (*Park et al., 2018*; *Ransick et al., 2019*). We successfully identified known cell-type-specific damage-induced genes such as *Havcr1* (kidney injury molecule-1, KIM1), *Krt8* (keratin 8), *Krt20* (keratin 20), and *Lcn2* (neutrophil gelatinase-associated lipocalin, NGAL) selectively in ischemia-reperfusion-injured (IRI) kidneys, but not in homeostatic uninjured control kidneys (*Figure 1—figure supplement 3C*), (*Liu et al., 2017*; *Ichimura et al., 2008*; *Paragas et al., 2011*).

Based on the cell clustering and gene expression patterns, we noticed that there are at least three epithelial cell states (homeostatic normal, activated, and dedifferentiated cells) in our dataset (See *Figure 1A*, right panel). Homeostatic normal cells express high expression of 'anchor' genes involved in normal cell function and identity (*Figure 1—figure supplement 3, B and C*). Most of the tubular epithelial cells from IRI kidneys robustly expressed damage-induced genes (ex. *Havcr1*, *Krt8*, *Krt20*, *Lcn2*), indicating they are in activated states (*Figure 1—figure supplement 3C*). These activated cells and homeostatic cells were grouped in the same cluster because they both highly express anchor genes characteristic for normal tubular epithelial states and functions (*Figure 1*, B and C; Fig, *Figure 1—figure supplement 3C*). However, we also identified additional damage-associated tubular epithelial clusters (*Figure 1C*, arrowheads; DA-PT, DA-TAL, and DA-DCT) that had lost or reduced expression of 'normal' mature epithelial cell marker genes but highly expressed damage-induced genes (*Figure 1—figure supplement 3, B and C*, *Figure 1A*).

Among these damage-associated epithelial cell clusters, we found a damage-associated proximal tubular cell state (See DA-PT cluster), which shows reduced homeostatic gene expression (ex. *Lrp2*, *Slc34a1*, *Hnf4a*, and *Acsm2*) and enrichment for genes associated with both renal development and kidney injury in human and mouse (ex. *Cdh6*, *Sox9*, *Sox4*, *Cited2*, *Vcam1*, *Vim*, and *Havcr1*; *Figure 1D*, and *Figure 1—figure supplement 5B* and *Figure 1—figure supplement 6A*), (*Combes et al., 2019a*; *Famulski et al., 2012*; *Adam et al., 2017*). Moreover, gene ontology enrichment analyses of this cellular population revealed proinflammatory molecular signatures and enriched expression of chemokines and cytokines such as *Cxcl2*, *Cxcl1*, *Ccl2*, and *Spp1* (*Figure 1—figure supplement 5B,D and E*). Reduced expression of *Hnf4a*, which is a transcription factor essential for the maturation of PT cells (*Marable et al., 2018*; *Marable et al., 2020*), and other homeostatic genes and upregulation of *Cdh6*, which is selectively expressed in immature proximal tubule progenitors in development and is essential for renal epithelialization (*Marable et al., 2020*; *Cho et al., 1998*; *Mah et al., 2000*), suggest that the cells in this cluster (DA-PT) are in a less differentiated cell state (*Figure 1D* and *Figure 1—figure supplement 6A*), (*Marable et al., 2020*). Then, we compared the transcriptional signature of this damage-associated PT cell state with previously published neonatal kidney single-cell RNA seq data (GSE94333, *Figure 1—figure supplement 7, A and B*), (*Adam et al., 2017*). The top 100 genes enriched in immature early PT cells in neonatal kidneys were mainly expressed in this damage-associated PT cell state (DA-PT, *Figure 1—figure supplement 7, C and D*). These analyses support our notion that the cells in the DA-PT cluster are in a dedifferentiated inflammatory state.

Among the damage-induced genes expressed in this dedifferentiated inflammatory PT cell state, we focused on the enrichment of *Sox9* and *Vcam1* (*Figure 1D*, See DA-PT cluster, arrowheads). Recent single-nucleus transcriptomic profiling of mouse IRI-kidneys identified vascular cell adhesion molecule 1 (VCAM1) as a marker of non-repairing proximal tubular cell state (*Kirita et al., 2020*), and *Vcam1* induction has been observed in multiple forms of human kidney diseases, including allograft rejection (*Hauser et al., 1997*). SRY-box9 (SOX9) is an essential transcription factor for successful renal repair after acute ischemic and toxic insults (*Kang et al., 2016*; *Kumar et al., 2015*) and is involved in the development of multiple organs, including mouse and human kidneys (*Reginensi et al., 2011*). SOX9 contributes to tissue repair processes by conferring stemness,

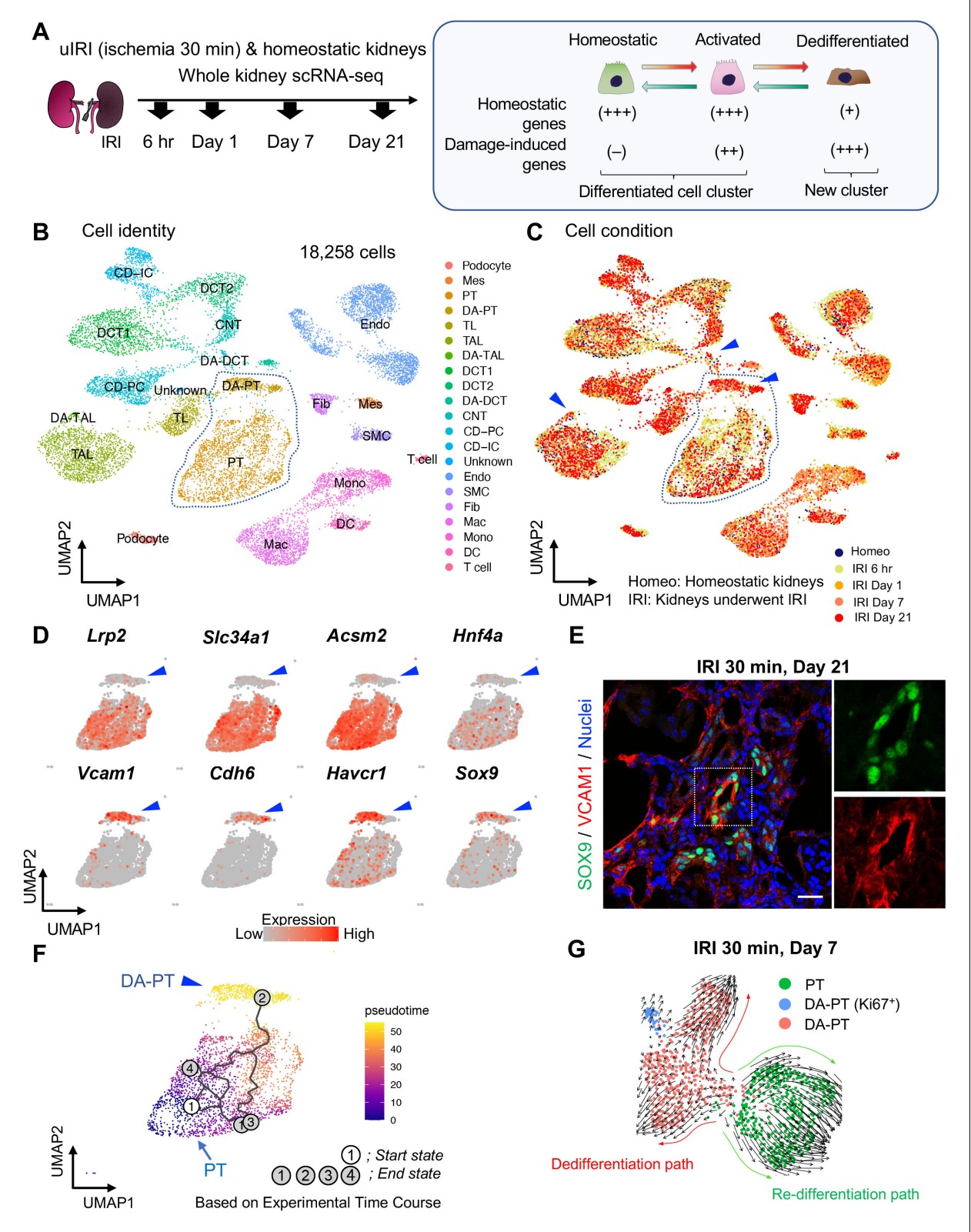

**Figure 1.** Single-cell RNA sequencing (scRNA-seq) identifies dynamic cellular state transitions of tubular epithelial cells after severe IRI. (**A**) Drop-seq strategy. uIRI, unilateral IRI. A schematic illustration of epithelial cell states is shown. (**B**) and (**C**) Integrated single-cell transcriptome map. Unsupervised clustering identified 21 distinct clusters in the UMAP plot. Arrowheads indicate damage-associated tubular epithelial cells. The dotted area (PT cell clusters; PT and DA-PT) was used for the downstream analyses in (**D**)–(**G**). (**D**) UMAP plots showing the expression of indicated genes in PT cell clusters

*Figure 1 continued on next page*

*Figure 1 continued*

(PT and DA-PT in (**B**)). Differentiated/mature PT cell markers: *Lrp2* (megalin), *Slc34a1* (sodium-dependent phosphate transporter 2a, NaPi2a), *Acsm2* (acyl-coenzyme A synthetase), and *Hnf4a* (hepatocyte nuclear factor 4α); and damage-induced genes: *Vcam1* (vascular adhesion molecule 1), *Cdh6* (cadherin 6), *Havcr1* (kidney injury molecule-1, KIM1), *Sox9* (Sry-box 9). Arrowheads; DA-PT. (**E**) Immunostaining for SOX9 and VCAM1 using post-severe IRI kidneys on day 21. Scale bar: 20 μm. (**F**) Pseudotime trajectory analysis of proximal tubular cells (PT and DA-PT clusters) that underwent IRI. A region occupied with cells from 6 hr after post-IRI was set as a starting state. (**G**) RNA velocity analysis of PT clusters (PT and DA-PT) from post-IRI kidneys on day 7. Cells in PT clusters from IRI day 7 dataset was extracted for the analysis. The arrows indicate predicted lineage trajectories. PT, proximal tubule; DA-PT, damage-associated PT; TL, thin limb; TAL, thick ascending limb; DA-TAL, damage-associated TAL; DCT, distal convoluted tubule; DA-DCT, damage-associated DCT; CNT, connecting tubule; CD, collecting duct (P, principal cells, IC, intercalated cells); Mes, mesangial cells; Endo, endothelial cells; SMC, smooth muscle cells; Fib, fibroblasts; Mac, macrophages; Mono, monocytes; DC, dendritic cells.

The online version of this article includes the following figure supplement(s) for figure 1:

**Figure supplement 1.** Characterization of severe and mild unilateral IRI models.
**Figure supplement 2.** Severe IRI leads to cystic and atrophic kidneys 6 months after severe IRI.
**Figure supplement 3.** scRNA-seq identifies major cell types in homeostatic and post-IRI kidneys.
**Figure supplement 4.** UMAP plots show the expression pattern of anchor genes in homeostatic and post-IRI kidneys.
**Figure supplement 5.** Damage-associated PT cells show an inflammatory transcriptional signature.
**Figure supplement 6.** Severe IRI reduces expressions of proximal tubular differentiation markers.
**Figure supplement 7.** Comparative analyses of damage-associated PT cells and neonatal proximal tubular cells.
**Figure supplement 8.** Trajectory analyses predict lineage hierarchy from differentiated mature PT cells to damage-associated PT cells.

plasticity, and regenerative capacity (*Kang et al., 2016*; *Kumar et al., 2015*; *Roche et al., 2015*; *Kadaja et al., 2014*; *Furuyama et al., 2011*; *Tata et al., 2018*). Our single-cell RNA-sequencing (scRNA-seq) data revealed that *Sox9* was most robustly induced in damage-associated PT cells compared to other tubular epithelial cells (DA-PT, *Figure 1—figure supplement 5C*). To validate this finding, we performed immunofluorescence for SOX9 and VCAM1 in histological sections of kidneys with failed repair. SOX9 nuclear accumulation was observed in VCAM1$^+$ proximal tubular cells (*Figure 1E*). High expression of *Sox9* and *Vcam1* suggests a potential role of this damage-associated PT cell state both in adaptive and maladaptive renal repair in a context-dependent manner, such as ranging severity of injury.

To understand the lineage hierarchy of PT cell states, we analyzed PT cells from differentiated and damage-associated PT cell clusters (PT and DA-PT in *Figure 1B*) using two algorithm tools (Monocle 3 and Velocyto) that allow the computational prediction of cell differentiation trajectories (*Cao et al., 2019*; *La Manno et al., 2018*). By placing each cell from the entire dataset in pseudotime we observed a predicted differentiation trajectory originating from PT to DA-PT (*Figure 1F* and *Figure 1—figure supplement 8, A and B*). We then performed RNA velocity analysis, which predicts the cell state trajectory based on the ratio between unspliced and spliced mRNA expressions, for these two PT cell states from the post-IRI dataset on day 7. Our RNA velocity analysis showed two trajectories running in opposite directions from the middle of the cluster, a position where genes associated with tubular maturation and damage are both not highly expressed (*Figure 1G* and *Figure 1—figure supplement 8C*). One projects toward the area with high levels of damage-induced genes (dedifferentiation path to damage-associated PT cell state) and the other toward the area with high levels of maturation-associated genes (redifferentiation path to differentiated PT cell state). Our computational analyses suggest the potential existence of cellular plasticity at this stage (Day 7 post-IRI; *Figure 1G* and *Figure 1—figure supplement 8C*).

## Proximal tubular cells dynamically alter their cellular states after acute kidney injury

To determine the temporal dynamics of damage-associated PT cell state in adaptive and maladaptive repair and validate the computational analyses, we performed expression analyses of multiple marker genes for this PT cell state in successful and failed renal repair processes. Quantitative RT-PCR analyses for *Sox9*, *Cdh6*, and *Vcam1* genes confirmed the transient induction of these genes and resolution after mild ischemic injury (20 min ischemia, Fig, 2B), but persistently elevated expression after severe ischemic injury (30 min) through 21 days after injury (*Figure 2B*). Using immunofluorescence and in situ hybridization, we observed more VCAM1$^+$ and *Cdh6*$^+$ tubular epithelial cells in IRI kidneys after 30 min than 20 min ischemia (*Figure 2C*). The number of SOX9-positive cells was

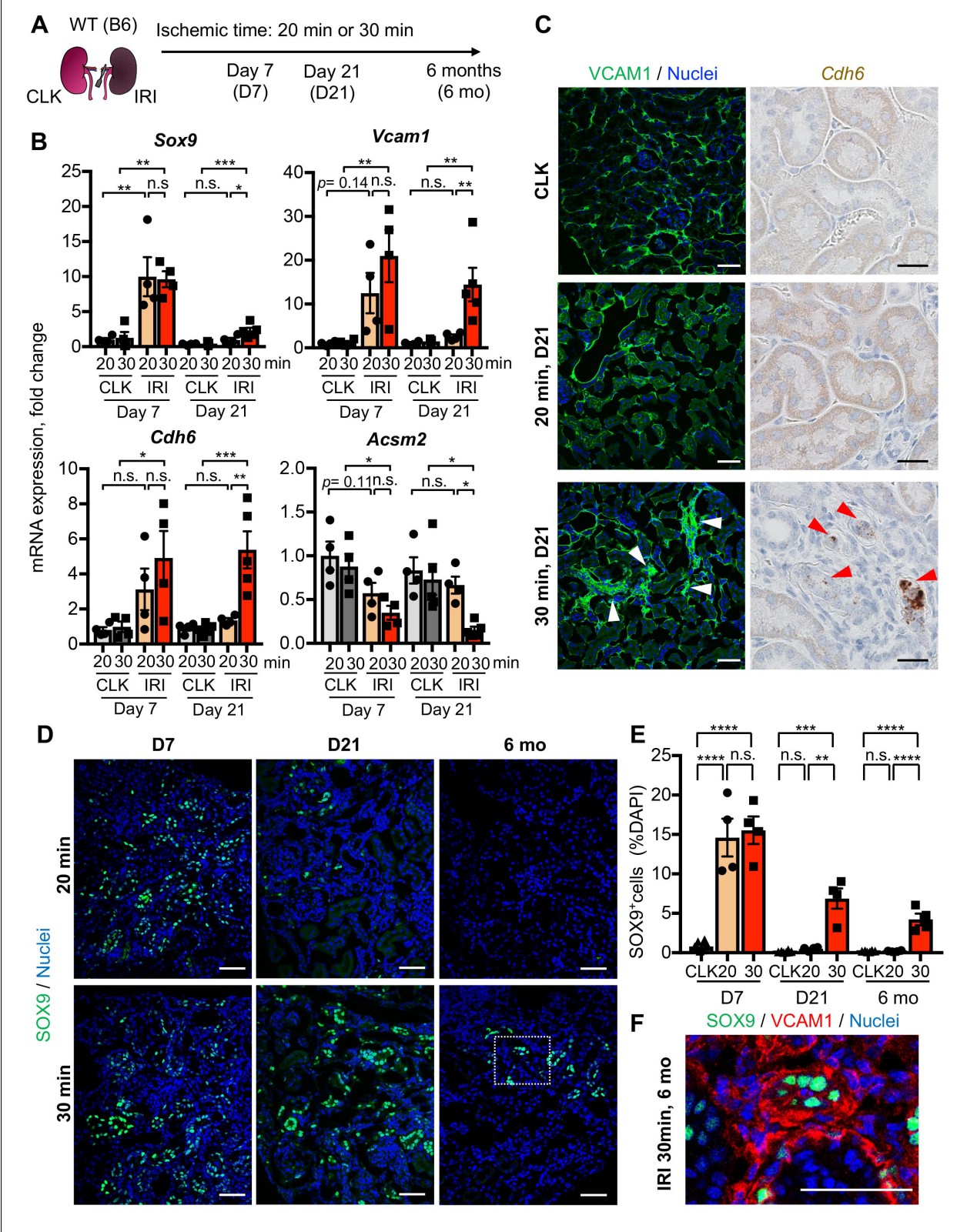

**Figure 2.** Damage-associated PT cells emerge transiently after mild injury but persist after severe injury. (**A**) Experimental workflow for the mild and severe IRI models. Left kidneys from wild-type (WT) C57BL/6J (B6) mice were subjected to mild (20 min) and severe (30 min) ischemia (unilateral IRI, uIRI). Contralateral kidneys (CLK) were used as controls. (**B**) Real-time PCR analyses of indicated gene expression. Whole kidney lysates were used. N = 4–5. (**C**) Expression analyses of VCAM1 and *Cdh6* using post-IRI kidneys on day 21. Immunostaining for VCAM1 revealed clusters of VCAM1^high tubular

*Figure 2 continued on next page*

*Figure 2 continued*

epithelial cells. In situ hybridization (ISH) was used to detect *Cdh6* gene expression on kidney sections. (**D**) Immunostaining for SOX9 in mild (20 min) and severe (30 min) IRI kidneys collected at indicated time points (day 7, day 21, and 6 months after IRI). (**E**) Quantification of SOX9⁺ cells over the DAPI⁺ area. Note that SOX9⁺ cells persist after severe IRI up to 6 months after IRI (30 for 30 min ischemia). In contrast, they disappear after a transient appearance in post-mild IRI kidneys (20 for 20 min ischemia). N = 4–8. (**F**) Immunostaining for SOX9 and VCAM1 (6 months post-severe IRI kidneys, dotted area in **D**). Scale bars, 20 μm in (**C**, *Cdh6*), and 50 μm (**C**, VCAM1, **D and F**). *p < 0.05; **p < 0.01; ***p < 0.001; ****p < 0.0001, one-way ANOVA with post hoc multiple comparisons test. n.s., not significant.

The online version of this article includes the following figure supplement(s) for figure 2:

**Figure supplement 1.** Comparative analyses of mild and severe IRI identify distinct temporal dynamics of damage-associated PT cells.

similarly increased between kidneys with mild and severe IRI on day 7 compared to baseline (*Figure 2*, D and E). Confocal imaging showed that most of the SOX9-positive cells co-express VCAM1 (*Figure 2—figure supplement 1, B and C*). Notably, SOX9 expression was reduced to baseline level in the kidneys that underwent mild injury while it persisted up to 6 months after severe IRI (*Figure 2*, D and E). We observed clusters of SOX9⁺VCAM1⁺ cells in the remaining parenchyma at 6 months post-severe IRI, but not in the post mild-IRI kidneys (*Figure 2F* and *Figure 2—figure supplement 1, B and C*). In accordance with the hypothesis that severe IRI injury is associated with increased signature of damage-associated PT cell state, there was a reduction of homeostatic gene expressions (*Acsm2* and *Slc34a1*) and the number of fully differentiated PT cells, which have high lotus tetragonolobus lectin (LTL)-binding (*Figure 2B* and *Figure 1—figure supplement 6, B-D*), (*Marable et al., 2020*). This finding is in line with a clinical correlation between low expression of *ACSM2B*, the human ortholog of *Acsm2*, and reduced renal function in patients with CKD (*Ledo et al., 2015*). Collectively, these data support the emergence and accumulation of damage-associated PT cells after severe injury but their return to a homeostatic state after mild injury.

To further characterize the dynamic changes and plasticity of proximal tubular cell state, we employed a *CreERT2* allele of *Sox9*, a highly enriched gene in the damage-associated PT cell state (DA-PT, *Figure 1—figure supplement 5C*), combined with *Rosa26^tdTomato* reporter to carry out lineage tracing (*Figure 3A*). In this mouse line (*Sox9^IRES-CreERT2*; *Rosa26^tdTomato*), tamoxifen administration permanently labels the *Sox9*-lineage cells with the tdTomato fluorescent reporter and provides the spatial information of the cells with a history of *Sox9* expression. On day 21, we found that severe ischemia (30 min) induces more robust accumulation of *Sox9*-lineage-labeled cells than mild ischemic injury (20 min) in the cortex and outer medulla of the IRI-kidneys (*Figure 3*, B and C). Approximately 25% percent of *Sox9*-lineage cells that underwent severe IRI were positive for VCAM1 on day 21, suggesting that part of *Sox9*-lineage cells did not fully redifferentiate after severe injury (*Figure 3*, D and E; 30 min). In contrast, only a few *Sox9*-lineage cells that underwent mild injury were VCAM1 positive at this time, indicating successful redifferentiation (*Figure 3*, D and E; 20 min). These results are consistent with the temporal dynamics of SOX9 and VCAM1 immunostaining results (*Figure 2—figure supplement 1*). Taken together, our data suggest that loss of plasticity and impaired redifferentiation of damage-associated PT cells underlie the failed renal repair/regeneration process (*Figure 3F*).

## Damage-associated PT cells create a proinflammatory milieu with renal myeloid cells

While an initial inflammatory response is critical for tissue repair, uncontrolled persistent inflammation underlies organ fibrosis (*Ferenbach and Bonventre, 2015*; *Humphreys, 2018*; *Gewin, 2018*). We hypothesized that the accumulation of damage-associated PT cells, which show proinflammatory transcriptional signature (*Figure 1—figure supplement 5B,D and E*), creates an uncontrolled inflammatory milieu by interacting with resident and infiltrating myeloid cells such as macrophages and monocytes (*Ide et al., 2020*). To determine the intercellular interactions between damage-associated PT cells and myeloid cells, we used NicheNet, a computational algorithm tool that infers ligand-receptor interactions and downstream target genes (*Figure 4*, A-D), (*Browaeys et al., 2020*). We applied NicheNet to predict ligand-receptor pairs in which ligands from damage-associated PT cells interact with receptors in monocyte or macrophages (*Figure 4*, A and C), (*Browaeys et al., 2020*). Among the top five predicted ligands expressed in damage-associated PT cells, we

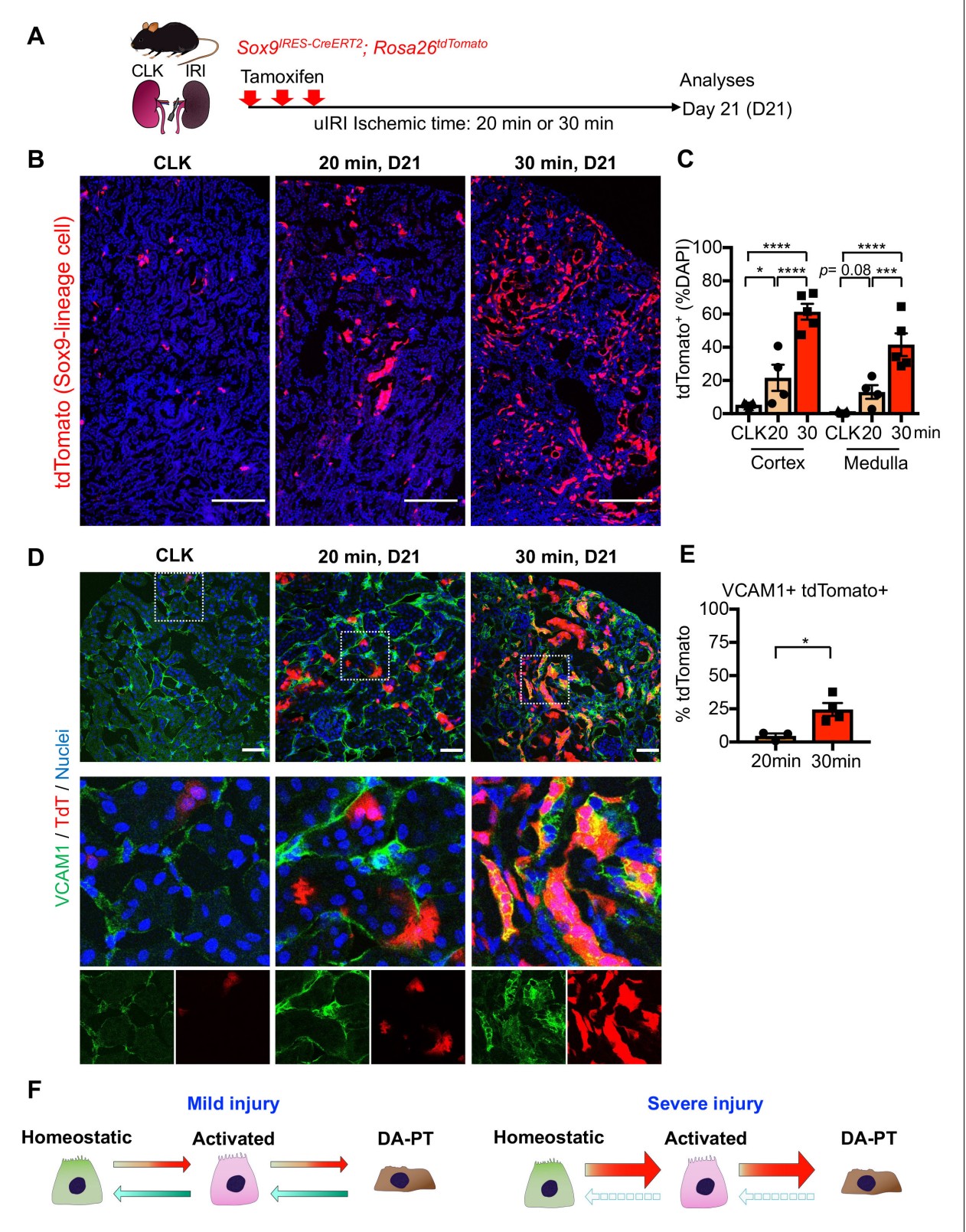

**Figure 3.** Lineage-tracing identifies the cellular plasticity of damage-associated PT cells. (**A**) Schematic of fate-mapping strategy using *Sox9^IRES-CreERT2*; *Rosa26^tdTomato* mice. Tamoxifen was administered three times on alternate days. Contralateral kidneys (CLK) were used as controls. (**B**) Distribution of tdTomato-expressing cells (*Sox9*-lineage cells) in contralateral (CLK), mild (20 min) and severe (30 min) IRI kidneys on day 21 (D21). (**C**) Quantification of tdTomato⁺ area relative to DAPI⁺ area in (**B**). DAPI was used for nuclear staining. N = 4–5. (**D**) Immunostaining for VCAM1 in *Sox9*-lineage-tagged
*Figure 3 continued on next page*

Figure 3 continued

kidneys (post-IRI, day 21). *Sox9*-lineage cells express native tdTomato red fluorescence (TdT). Insets: individual fluorescence channels. (E) Quantification of double-positive cells in total tdTomato+ cells in (D). N = 3–4. Note that more *Sox9*-lineage cells express VCAM1 after severe IRI (30 min) compared to mild IRI (20 min) on day 21. One-way ANOVA with post hoc multiple comparisons test and unpaired Student's t-test were used for (C) and (E), respectively. Scale bars, 200 μm in (B); and 50 μm in (D). *p < 0.05; **p < 0.01; ***p < 0.001; ****p < 0.0001. (F) Schematic illustration of PT cell state dynamics. Differentiated/mature PT cells are activated, transit into a molecularly distinct PT cell state (damage-associated PT cells in DA-PT cluster), and redifferentiate into their original state after mild injury (left). Severe injury prevents the redifferentiation of damage-associated PT cells into normal PT cell state, leading to the accumulation and persistence of damage-associated PT cells (right).

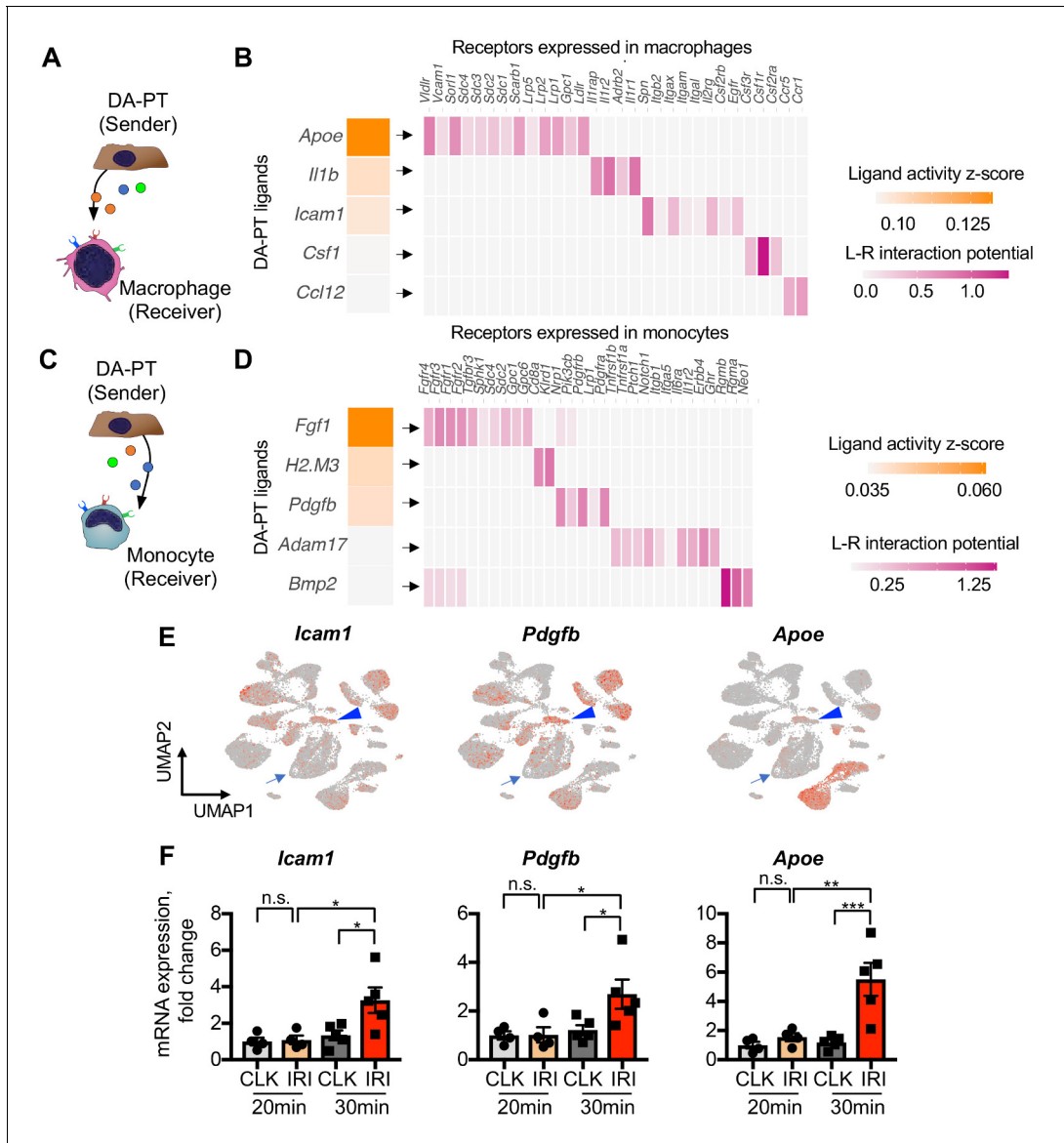

**Figure 4.** Damage-associated PT cells create a proinflammatory milieu with myeloid cells. (A) Schematic model of intercellular communications between damage-associated PT cells and macrophages. NicheNet was used to predict intercellular interactions using our integrated single-cell map of failed renal repair. (B) Predicted ligands from damage-associated PT cells and receptors in macrophages. (C) Schematic model of intercellular communications between damage-associated PT cells and monocytes. (D) Predicted ligands from damage-associated PT cells and receptors in monocytes. (E) UMAP plots showing the expression of indicated genes. Our integrated single-cell map of mouse failed renal repair is shown (See *Figure 1*, B and C). Arrowheads indicate damage-associated PT cells (DA-PT cluster). Arrows indicate differentiated PT cells (PT cluster). (F) Real-time PCR analyses of indicated gene expression. Post-IRI kidneys on day 21 that underwent mild (20 min) or severe (30 min) ischemia were used. N = 4–5. *p < 0.05; **p < 0.01; ***p < 0.001, one-way ANOVA with post hoc multiple comparisons test.

confirmed the enrichment of *Icam1*, *Pdgfb*, and *Apoe* expression in this cell state (*Figure 4E*, arrowheads). *Icam1* and *Pdgfb* have been implicated in human AKI (*Famulski et al., 2012*), and *Apoe* genetic variation has been linked with CKD progression (*Hsu et al., 2005*). As inferred by NicheNet, mRNA expression of *Icam1*, *Pdgfb*, and *Apoe* were markedly increased in the kidneys showing the accumulation of damage-associated PT cells compared to post-IRI kidneys without the accumulation (*Figure 4F*; 30 min *vs.* 20 min ischemia). These data delineate a complex inflammatory circuit within the damaged kidneys involving intercellular communication between damage-associated PT cells and myeloid cells that contribute to maladaptive renal repair.

## Damage-associated PT cells exhibit high ferroptotic stress after severe IRI

Next, we investigated the molecular mechanisms that are critical for cells to traverse between differentiated PT cells and damage-associated PT cells. To this end, we analyzed the transcriptional signature of PT cells in the differentiated/mature cluster to identify critical pathways to maintain this cellular state. We found that genes associated with glutathione metabolic processes and anti-oxidative stress response pathways are overrepresented in the differentiated mouse PT cell cluster (*Figure 5A*; *Figure 1—figure supplement 5F* and *Figure 5—figure supplement 1*). We also found that these pathways are enriched in normal differentiated human PT cells (*Figure 5A* and *Figure 5—figure supplement 2, A and B*; GSE131882). Mirroring these findings, oxidative stress-induced signaling pathways related to failed renal repair, such as cellular senescence and DNA damage responses (*Kishi et al., 2019*; *Canaud et al., 2019*), were highly enriched in damage-associated PT cells (*Figure 5—figure supplement 1*). Taken together, we propose that glutathione-mediated anti-oxidative stress responses are critical for maintaining the cellular identity of fully differentiated PT cells, and dysregulation of these pathways underlies the failure of damage-associated PT cells to redifferentiate into normal PT cell state.

Among the cellular stress pathways related to dysregulation of glutathione metabolism, ferroptotic stress and ferroptosis have been implicated in failed repair of human AKI and pathogenesis in mouse models of AKI, (*Figure 5B*), (*Stockwell et al., 2017*; *Dixon et al., 2012*; *Yang et al., 2014*; *Linkermann et al., 2014*; *Wenzel et al., 2017*; *Müller et al., 2017*). To investigate whether ferroptotic stress underlies the emergence and accumulation of damage-associated PT cells in addition to its known role in inducing cell death during maladaptive repair, we first tested the expression of the canonical anti-ferroptosis defense pathway, glutathione/GPX4 axis (*Figure 5B*). In agreement with the underrepresentation of glutathione metabolic process in damage-associated PT cells, the genes encoding the glutathione/GPX4 defense pathway were markedly downregulated in this PT cell state (DA-PT) compared to differentiated PT cells (PT), suggesting that damage-associated PT cells are potentially vulnerable to ferroptotic stress (*Figure 5C*).

We then analyzed the expression of ferroptotic stress biomarkers such as malondialdehyde (MDA, a lipid peroxidation product) and acyl-CoA synthetase long-chain family member 4 (ACSL4), which also regulate cellular sensitivity to ferroptosis (*Kagan et al., 2017*; *Doll et al., 2017*; *Müller et al., 2017*; *Kenny et al., 2019*; *Yuan et al., 2016*; *Li et al., 2019*). A recent pharmacological inhibitor study showed that ACSL4 is a reliable maker for ferroptotic stress in murine model of ischemic AKI (*Zhao et al., 2020*). We identified significant upregulation of *Acsl4* in damage-associated PT cells in dot-plots (*Figure 5C*). The co-expression of markers for damage-associated PT cells and ferroptotic stress was confirmed by immunofluorescence for SOX9, MDA, and ACSL4 (*Figure 5, D-G*). We found that severe ischemia (30 min) induces more expression of ferroptotic stress markers in SOX9$^+$ cells than mild ischemic injury (20 min) (*Figure 5, E and G*). These data demonstrate that SOX9$^+$ damage-associated PT cells undergo high ferroptotic stress after severe ischemic injury.

To address whether the emergence of damage-associated PT cells is specific to IRI injury or appears in other cases of acute kidney injury, we investigated the co-expression of SOX9 and VCAM1 in models of toxic renal injury (aristolochic acid nephropathy, AAN) and obstructive renal injury (unilateral ureteral obstruction, UUO), which lead to severe fibrosis. By immunofluorescence analyses of SOX9 and VCAM1 co-expression, we found the emergence of damage-associated PT cells in both models (*Figure 6, A and C*). Furthermore, the SOX9-positive tubular epithelial cells in these models showed co-expression of ACSL4, suggesting that ferroptotic stress of damage-associated PT cells is a conserved response to kidney injury across various etiologies (*Figure 6, B and C*).

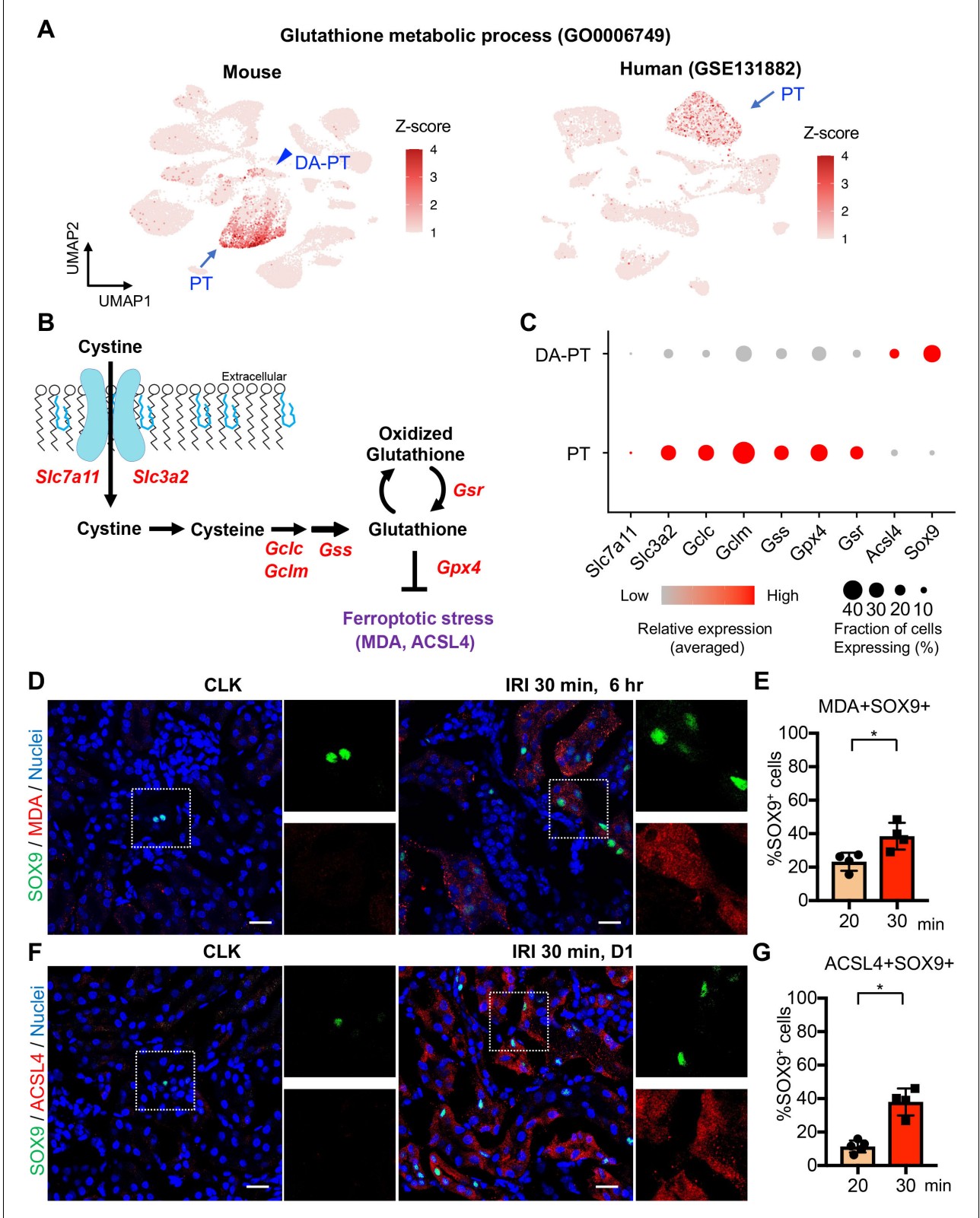

**Figure 5.** Damage-associated PT cells undergo high ferroptotic stress after severe IRI. (**A**) UMAP rendering of glutathione metabolic process in mouse and human kidneys. (**B**) A scheme showing glutathione-glutathione peroxidase 4 (GPX4) anti-ferroptotic defense pathway. *Slc7a11* and *Slc3a2* (system x$_c^-$); *Gclc* and *Gclm* (glutamate-cysteine ligase); *Gss* (glutathione synthetase); *Gsr* (glutathione reductase): and *Gpx4*. MDA (malondialdehyde, a lipid peroxidation product) and ACSL4 (acyl-CoA synthetase long-chain family member 4) are markers for ferroptotic stress. (**C**) Dot plots show the

*Figure 5 continued on next page*

*Figure 5 continued*

expression of genes for glutathione-GPX4 axis, *Sox9,* and *Acsl4.* (**D**) Immunostaining for SOX9 and MDA (6 hr post-IRI), and (**E**) quantification of double-positive cells in total SOX9$^+$ cells. N = 4. (**F**) Immunostaining for SOX9 and ACSL4 (1 day post-IRI), and (**G**) quantification of double-positive cells in total SOX9$^+$ cells. N = 4. Insets: individual fluorescence channels of the dotted box area. Note that severe ischemia (30 min) induces more ferroptotic stress markers (MDA and ACSL4) in SOX9$^+$ cells in damaged kidneys than mild ischemia (20 min). Wild-type C57BL/6J mice were used for (**D**) to (**G**). Scale bars, 20 μm in (**D**) and (**F**). *p < 0.05. unpaired Student's t-test.

The online version of this article includes the following figure supplement(s) for figure 5:

**Figure supplement 1.** Gene ontology analyses identify enrichment of anti-oxidative stress defense genes in differentiated/mature PT cells.
**Figure supplement 2.** Characterization of human normal kidney single-nucleus RNA-seq data.

We then investigated whether molecularly similar damage-associated PT cells can be observed in human AKI. We analyzed scRNA-seq data from biopsy samples of two transplanted human kidneys with evidence of AKI and acute tubular injury but no evidence of rejection (GSE145927; *Figure 6D* and *Figure 6—figure supplement 1A*), (*Malone et al., 2020*). We found a cell population that is enriched for genes expressed in mouse damage-associated PT cells, including *SOX9, VCAM1, CDH6, and VIM*. This cellular population also showed decreased expression of homeostatic PT genes (*ALDOB, MIOX,* and *GPX4*) (*Figure 6*, State 3; D; and E). Trajectory inference using Monocle 3 suggests that damage-associated PT cells emerge from mature differentiated PT cells with high expression of homeostatic genes in human kidneys (PT to DA-PT-like in *Figure 6F*). Interestingly, the glutathione metabolic gene signature is high in mature PT cells and decreases along the trajectory to DA-PT-like cells (*Figure 6—figure supplement 1C*). These data suggest that the emergence of damage-associated PT cells is a mechanism of acute kidney injury and repair that is shared by humans and mice.

## Genetic induction of ferroptotic stress results in accumulation of inflammatory PT cells after mild injury

Our data suggest that severe injury, which induces more oxidative and ferroptotic stress than mild injury, causes the accumulation of inflammatory damage-associated PT cells and worsens long-term renal outcomes. We hypothesized that ferroptotic stress plays a crucial role in driving the accumulation of inflammatory PT cells and promoting maladaptive repair in addition to triggering cell death (ferroptosis). To test this hypothesis, we generated a mouse model that selectively and conditionally deletes *Gpx4* in *Sox9*-lineage cells (*Sox9$^{IRES-CreERT2}$; Gpx4$^{flox/flox}$*, hereafter conditional knockout [cKO]; *Figure 7A*). Genetic deletion of *Gpx4* robustly induces ferroptotic stress and triggers ferroptosis (*Yang et al., 2014*; *Friedmann Angeli et al., 2014*). In this mouse line, exons 4–7 of the *Gpx4* allele, which include the catalytically active selenocysteine site of the GPX4 protein, is deleted in a tamoxifen-inducible manner selectively in *Sox9*-lineage cells. We subjected the cKO mice and littermate control mice to mild renal ischemic stress (ischemic time 22 min). This condition induces robust *Sox9-CreERT2* expression but does not induce the failed renal repair phenotype in control mice (*Gpx4* $^{flox/flox}$). We induced *Gpx4* deletion at the time of injury by tamoxifen injection (*Figure 7A*). The littermate control mice were subjected to the same renal ischemic stress and tamoxifen. We confirmed successful deletion of GPX4 protein by immunofluorescence (*Figure 7—figure supplement 1, B and C*) and found that expression of the ferroptotic stress marker ACSL4 was increased on day 21 post-IRI in cKO kidneys compared to littermate kidneys that underwent the same ischemic stress (*Figure 7—figure supplement 1, D and E*). Contralateral uninjured kidneys from cKO mice only showed a minimum deletion of GPX4 as the *Sox9-CreERT2* activity is not induced in non-injured proximal tubular cells (See *Figure 3B* for CLK), (*Kumar et al., 2015*).

The post-ischemic cKO kidneys were atrophic and showed severe tubular injury on histological evaluation on day 21 and exhibited marked accumulation of KIM1$^+$KRT8$^+$ injured tubular cells (*Figure 7*, B-D and *Figure 7—figure supplement 2*). By contrast, control littermate kidneys that underwent the same ischemic stress exhibited resolution of histological changes and fewer KIM1$^+$KRT8$^+$ cells (*Figure 7*, C and D, and *Figure 7—figure supplement 2*). Contralateral kidneys from both genotypes showed neither increased KIM1 nor KRT8 expression (*Figure 7—figure supplement 3, A and C*). The post-ischemic cKO kidneys also exhibited massive accumulation of F4/80$^+$ macrophages, αSMA$^+$ myofibroblasts, and increased collagen synthesis (*Figure 7*, E-F; *Figure 7—figure*

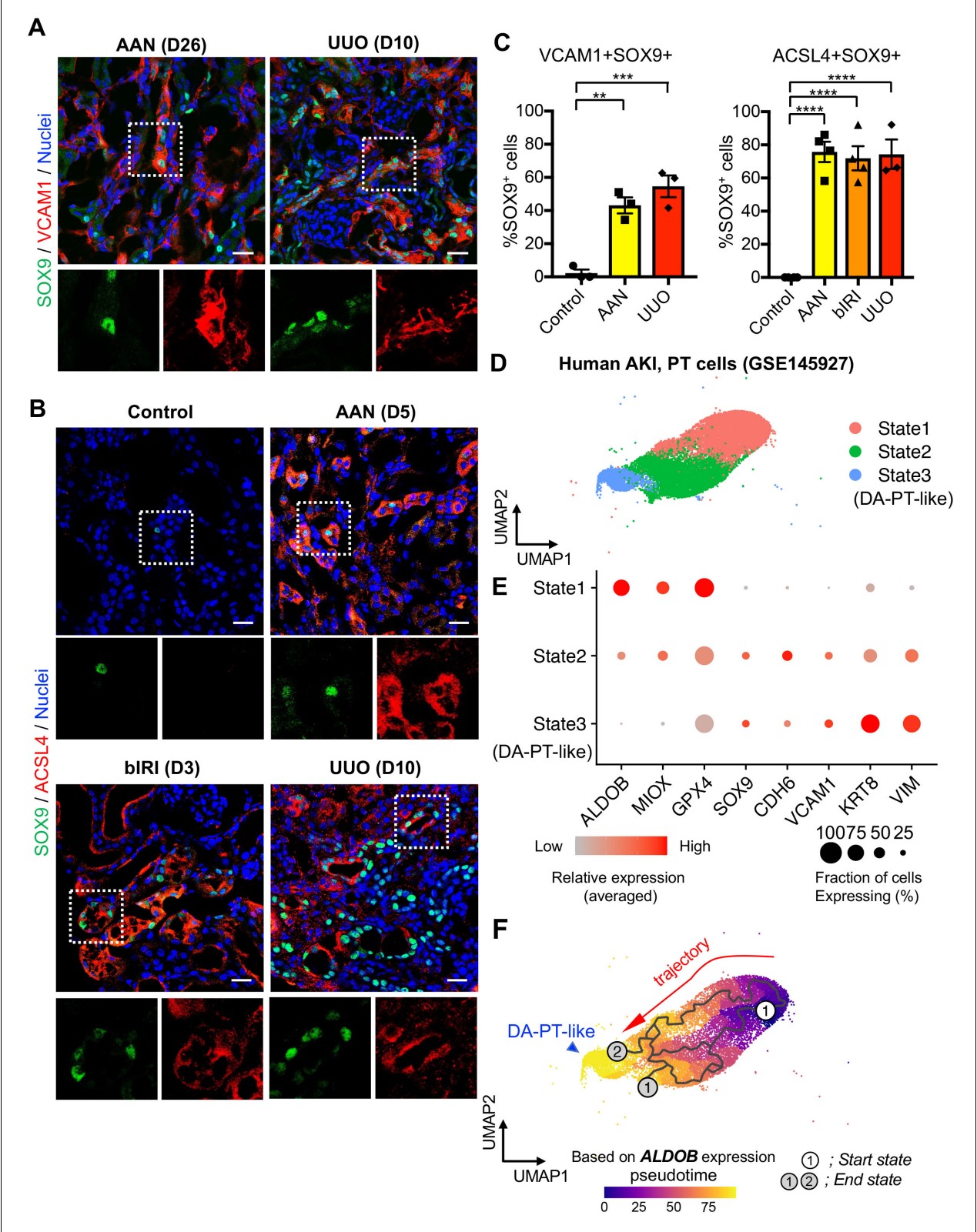

**Figure 6.** Damage-associated PT cells emerge after injury in mouse and human kidneys. (**A**) Immunostaining for SOX9 and VCAM1. Aristolochic acid nephropathy (AAN) and unilateral ureteral obstruction (UUO) models were used. Kidneys from wild-type C57BL/6J mice were harvested on day 26 (D26) for AAN and day 10 (D10) for UUO. Insets: individual fluorescence channels of the dotted box area. (**B**) Immunostaining for SOX9 and ACSL4. bIRI, bilateral IRI model. Kidneys were harvested on day 3 (D3) for bIRI, day 5 (D5) for AAN, and day 10 (D10) for UUO. Insets: individual fluorescence

*Figure 6 continued on next page*

*Figure 6 continued*

channels of the dotted box area. (C) Quantification of double-positive cells in total SOX9$^+$ cells from panel (A) and (B). Scale bars, 20 μm. N = 3–4. **p < 0.01; ***p < 0.001; ****p < 0.0001, one-way ANOVA with post hoc multiple comparisons test. (D) UMAP of the human proximal tubular cells from AKI kidneys. (E) Dot plots showing the expression of indicated genes. Note that PT cells in state 3 (DA-PT-like) show increased gene expressions of markers for mouse damage-associated PT cells (*SOX9, VCAM1, CDH6*) and reduced expression of homeostatic genes (*ALDOB, MIOX, and GPX4*). (F) Pseudotime trajectory analysis of PT clusters (PT and DA-PT-like cells). A region occupied with *ALDOB*$^{high}$ cells were set as a starting state. Arrow, predicted trajectory from PT cells to DA-PT-like cells.

The online version of this article includes the following figure supplement(s) for figure 6:

**Figure supplement 1.** Characterization of human AKI kidney single-cell RNA-seq data.

---

*supplement 3, B and C*). Then, we assessed the number of cell death by terminal deoxynucleotidyl transferase-mediated dUTP nick end labeling (TUNEL) assay, which detects ferroptotic cell death in *Gpx4*-deleted tissues (*Friedmann Angeli et al., 2014*). Consistent with the known role of GPX4 to prevent ferroptosis, genetic deletion of *Gpx4* led to the increased TUNEL$^+$ tubular epithelial cells in cKO kidneys (*Figure 7*, G and H; See *Figure 7—figure supplement 4* for CLK). Collectively, these data indicate that genetic induction of ferroptotic stress in *Sox9*-lineage cells is sufficient to prevent normal renal repair after mild ischemic injury and to mimic the failed renal repair phenotype observed after severe ischemic injury.

We then investigated if the number of damage-associated PT cells was increased in the *Gpx4* cKO kidneys after mild ischemic injury. While VCAM1 is strongly induced in damage-associated PT cells and serves as a reliable marker, it is also expressed weakly in F4/80$^+$ macrophages and endo-mucin (EMCN)$^+$ endothelial cells after kidney injury (*Figure 1—figure supplement 6A*; see UMAP). For the precise quantification of damage-associated PT cells, we co-stained the kidneys with VCAM1, EMCN, and F4/80, and scored VCAM1$^+$F4/80$^-$EMCN$^-$ cells as damage-associated PT cells (*Figure 8*, B and C; and *Figure 7—figure supplement 3D*). Supporting our hypothesis, we observed increased numbers of VCAM1$^+$EMCN$^-$F4/80$^-$ cells in post-ischemic cKO kidneys on day 21, while the value was at a baseline level in control littermate kidneys that underwent the same mild ischemic stress (*Figure 8*, B and C). We further employed a genetic fate-mapping strategy in *Gpx4*-deficient *Sox9*-lineage cells by generating a mouse line that harbors *Sox9*$^{IRES-CreERT2}$; *Gpx4*$^{flox/flox}$; *Rosa26*$^{tdTomato}$ alleles. Confocal imaging identified the colocalization of tdTomato (*Sox9*-lineage) and VCAM1 and ACSL4 in the post-IRI cKO kidneys (*Figure 8D*). Other molecular markers of damage-associated PT cell state, such as *Cdh6* and *Sox9*, were also increased in cKO kidneys on day 21 post-IRI (*Figure 8E*). These VCAM1$^+$ cells were also positive for SOX9 (*Figure 8F*). In situ hybridization confirmed robust *Cdh6* expression in tubular epithelial cells in post-ischemic cKO kidneys (*Figure 8G* and *Figure 7—figure supplement 3E*). These data substantiate our model that ferroptotic stress drives the accumulation of damage-associated PT cells by preventing redifferentiation of these transient inflammatory epithelial cells into normal PT cell state and augments renal inflammation and fibrosis (*Figure 8H*).

## Pharmacological inhibition of ferroptotic stress prevents the accumulation of inflammatory PT cells and ferroptosis after ischemia-reperfusion injury

We next investigated whether pharmacological inhibition of ferroptosis blunts the dynamic changes seen in proximal tubular cells. We administered liproxstatin-1, an in vivo active ferroptosis inhibitor that scavenges lipid peroxides (*Friedmann Angeli et al., 2014*), to our cKO mice that underwent mild renal ischemia (*Figure 9A*). The same volume of vehicle solution (1% dimethyl sulfoxide in phosphate-buffered saline) was administered to cKO and littermate controls (*Gpx4*$^{flox/flox}$), and these animals underwent the same procedure of unilateral IRI. While vehicle-treated IRI-kidneys of control genotype did not show renal atrophy, cKO IRI-kidneys with daily vehicle injections exhibited renal atrophy (*Figure 9B*). We further confirmed effective genetic targeting in our cKO IRI-kidneys from the vehicle and liproxstatin-1-treated groups by using tdTomato-lineage tracing and GPX4 immunohistochemistry (*Figure 9—figure supplement 1, B and C*). Daily administration of liproxstatin-1 potently mitigated the renal atrophy and reduced expression of renal tubular injury markers in cKO IRI-kidneys (KIM1 and KRT8; *Figure 9*, B-D; *Figure 9—figure supplement 2*). Notably, liproxstatin-1

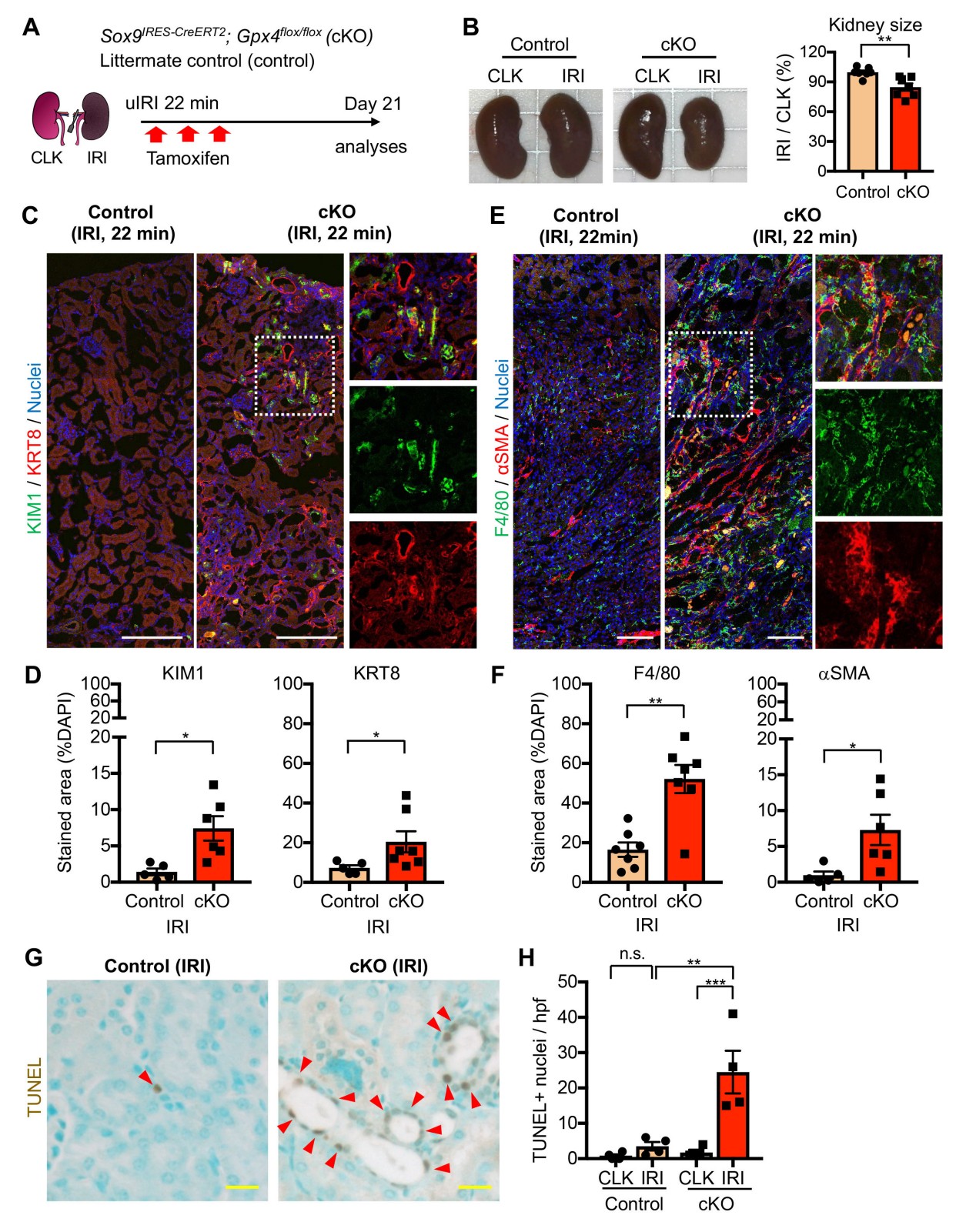

**Figure 7.** Genetic induction of ferroptotic stress to *Sox9*-lineage cells augments kidney injury. (**A**) Experimental workflow for *Gpx4* deletion in *Sox9*-lineage cells. uIRI, unilateral IRI (ischemic time 22 min). Kidneys were harvested on day 21 post-IRI. cKO mice and their littermate controls were subjected to the same ischemic stress and tamoxifen treatment. *Gpx4* is deleted in *Sox9*-lineage cells after IRI with tamoxifen administration. (**B**) The deletion of *Gpx4* results in renal atrophy. Relative size of post-IRI kidneys compared to contralateral kidneys (CLK) was quantified. Control, littermate

*Figure 7 continued on next page*

*Figure 7 continued*

control. N = 7. (C) and (D) Immunostaining for tubular injury markers (KIM1 and KRT8). IRI kidneys from cKO and control littermates are shown. CLK did not show KIM1 or KRT8 staining. Quantification of KIM1 or KRT8-positive area over the DAPI$^+$ area is shown in (D). N = 5–7. (E) and (F) Immunostaining for F4/80 and αSMA. IRI kidneys from cKO and control littermates are shown. Quantification of F4/80 or αSMA-positive area over the DAPI$^+$ area is shown in (F). N = 5–7. Insets: individual fluorescence channels of the dotted box area. (G) and (H) TUNEL staining for evaluating cell death. Quantification of TUNEL-positive nuclei is shown in (H). N = 4. Arrowheads, TUNEL$^+$ nuclei. Abbrev: hpf, high power field. Unpaired t-test for (D) and (F). One-way ANOVA with post hoc multiple comparison test for (H). Scale bars, 100 μm in (C) and (E); 20 μm in (G).

The online version of this article includes the following figure supplement(s) for figure 7:

**Figure supplement 1.** Genetic deletion of *Gpx4* leads to augmented ferroptotic stress after mild IRI.
**Figure supplement 2.** Genetic deletion of *Gpx4* leads to severe kidney injury after mild IRI.
**Figure supplement 3.** Genetic deletion of *Gpx4* leads to the accumulation of damage-associated PT cells.
**Figure supplement 4.** Genetic deletion of *Gpx4* leads to cell death of tubular epithelial cells.

prevented the accumulation of SOX9+VCAM1+ proximal tubular cells (*Figure 9*, E and F). Quantitative RT-PCR analyses confirmed that the expression of *Sox9*, *Vcam1*, and *Cdh6* (DA-PT markers) were all reduced by liproxstatin-1 to the same level as in contralateral uninjured kidneys (*Figure 9G*). Moreover, TUNEL staining showed a significant reduction of TUNEL+ cells in the liproxstatin-1-treated IRI-cKO kidneys compared to the vehicle-treated IRI-cKO kidneys (*Figure 9*, H and I). Collectively, liproxstatin-1 potently ameliorated the pathologic changes of proximal tubular cells and overall damage of *Gpx4*-deficient kidneys that underwent IRI (*Figure 9J*).

## Discussion

By using complementary scRNA-seq and mouse genetic approaches in several experimental models of renal injury and repair, our study revealed novel mechanisms regulating proximal tubular cell states that underlie renal repair and regeneration. By detailed characterization of damage-associated PT cells in our single-cell map of failed repair, we identified that this PT state significantly downregulates the canonical anti-ferroptosis defense pathway, making them potentially vulnerable to ferroptotic stress. Genetic induction of ferroptotic stress after mild injury was sufficient to prevent the redifferentiation of damage-associated PT cells into the normal PT cell state, leading to the accumulation and persistence of inflammatory PT cells that promote maladaptive repair. Our data collectively advances our understanding of the ferroptotic cell death pathway by identifying a novel role of ferroptotic stress in promoting and accumulating pathologic cellular state beyond its known role to trigger non-apoptotic regulated cell death (ferroptosis). GPX4 is a key coordinator of proximal tubular cell fate for renal repair and regeneration by preventing both cell death and cell death-independent pathologic changes after IRI.

Unbiased clustering of cells clearly separates damage-associated PT cells from homeostatic and activated differentiated PT cells, indicating that damage-associated PT cells represent a unique cellular status. We also found a molecularly similar PT cell state in kidneys of patients with acute kidney injury. Similar to our current findings, we and others have identified the emergence of molecularly distinct epithelial cells during the process of lung injury and repair (*Kobayashi et al., 2020*; *Choi et al., 2020*; *Strunz et al., 2020*). These novel transient cells are termed as pre-alveolar type-1 transitional cell state (PATS), alveolar differentiation intermediate, and damage-associated transient progenitors. They originate from alveolar type two epithelial cells and differentiate into type one alveolar epithelial cells (*Kobayashi et al., 2020*; *Choi et al., 2020*; *Strunz et al., 2020*). PATS and PATS-like cells in humans accumulate during failed lung repair and fibrosis (*Kobayashi et al., 2020*), as in the case of maladaptive repair of kidneys. Molecular mechanisms underlying the accumulation of these transitional cell state include hypoxia, inflammation, and DNA damage. All these pathways promote maladaptive renal repair by altering PT cell states (*Strausser et al., 2018*; *Liu et al., 2017*; *Ferenbach and Bonventre, 2015*; *Kishi et al., 2019*). These data suggest that the emergence of molecularly distinct epithelial cell states and their persistence/accumulation is a general mechanism of maladaptive repair in multiple organs across mice and humans.

The complexity of proximal tubular cell states in renal injury and repair processes has been recently identified at single-cell resolution (*Kirita et al., 2020*; *Rudman-Melnick et al., 2020*). A recent study investigated PT cellular heterogeneity using single-nucleus RNA sequencing in a mouse

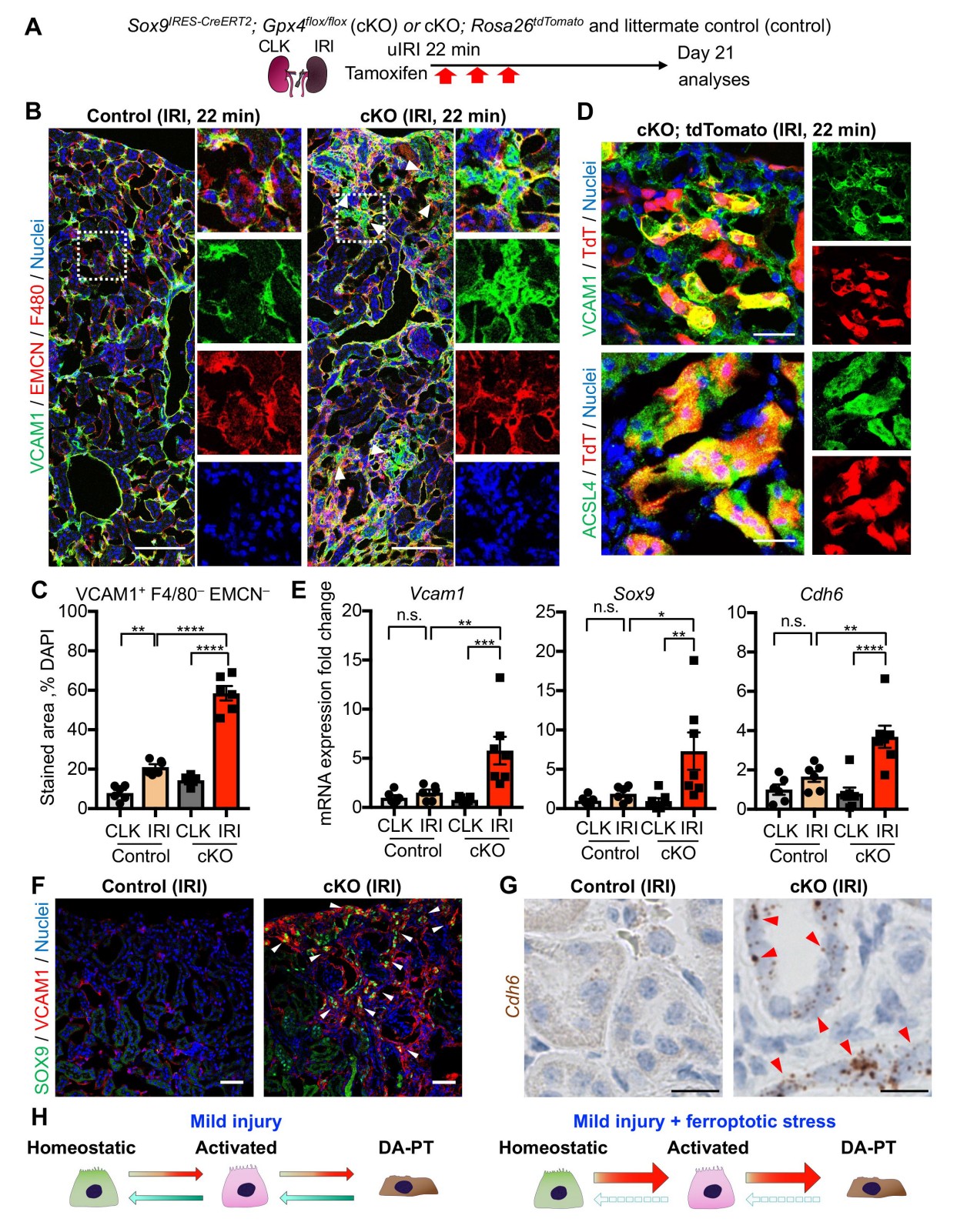

**Figure 8.** Genetic induction of ferroptotic stress induces the accumulation of damage-associated PT cells after mild injury. (**A**) Schematic representation of experimental workflow. tdTomato-lineage tracing was employed to detect *Sox9*-lineage cells. cKO mice and their littermate controls were subjected to the same ischemic stress (ischemic time, 22 min) and tamoxifen treatment. Kidneys were harvested on day 21 post-IRI. (**B**) and (**C**) Immunostaining for VCAM1, EMCN (endomucin), and F4/80. IRI kidneys from cKO and control littermates (control) are shown. Quantification of VCAM1+EMCN−F4/80− area

*Figure 8 continued on next page*

**Figure 8 continued**

over the DAPI$^+$ area is shown in (**C**). N = 6. (**D**) Immunostaining for VCAM1, ACSL4, and native tdTomato (TdT) fluorescence. Insets: individual fluorescence channels. (**E**) Real-time PCR analyses of indicated gene expression. Whole kidney lysates were used. N = 6–7. *p < 0.05; **p < 0.01; ***p < 0.001; ****p < 0.0001, one-way ANOVA with post hoc multiple comparisons test for (**C**) and (**E**). (**F**) Immunostaining for SOX9 and VCAM1. Arrowheads indicate double-positive cells (damage-associated PT cells). (**G**) ISH for *Cdh6* expression. Red arrowheads indicate *Cdh6*-positive renal tubular cells. Scale bars, 100 μm in (**B**); 20 μm in (**D**); 50 μm in (**F**); and 10 μm in (**G**). (**H**) Schematic illustration of PT cell state dynamics. Differentiated/mature PT cells are activated, transit into a damage-associated inflammatory PT cell state (DA-PT), and redifferentiate to their original state after mild injury. Ferroptotic stress prevents the redifferentiation of damage-associated PT cells into normal PT cell state, leading to the accumulation of the pathologic PT cells that actively produce inflammatory signals.

model of bilateral renal IRI. The paper revealed multiple novel PT cellular states, ranging from severely injured cells, cells repairing from injury, and cells undergoing failed repair (*Kirita et al., 2020*). Interestingly, the damage-associated PT cells reported here shares some of the transcriptional signatures with so-called failed repair proximal tubular cells (FR-PTC), such as *Vcam1, Cp, Akap12,* and *Dcdc2a* among the Top 20 transcriptional signature of FR-PTC. In contrast, we also found some differences between the damage-associated PT cells and FR-PTC. The most highly expressed genes in FR-PTC (ex. *Kcnip4, Dock10, Pdgfd, Erbb4,* and *Psd3*) were not expressed in damage-associated PT cells and vice versa. Moreover, damage-associated PT cells act like a transient cell state. They redifferentiate to the homeostatic PT cell state after mild injury while they accumulate after severe injury. Damage-associated PT cells may represent a broad transient cell state, including FR-PTC.

Another study profiled juvenile (4-week-old) mouse kidneys that underwent 30 min unilateral IRI (*Rudman-Melnick et al., 2020*). Unlike adult kidneys, the kidneys at this stage showed marked regenerative ability and showed successful repair. The study found transient induction of nephrogenic transcriptional signature (ex. *Sox4, Cd24a, Npnt, Lhx1, Osr2, Foxc1, Hes1, Pou3f3,* and *Sox9*) in damaged PT cells during the injury–repair process (*Rudman-Melnick et al., 2020*). While our transcriptional analyses of damage-associated PT cells indicate they are in a dedifferentiated state, they do not show a nephrogenic signature at the level of damaged juvenile kidneys (i.e. damage-associated PT cells were positive for *Sox4, Npnt,* and *Cd24a,* and *Sox9,* but negative for *Lhx1, Osr2, Foxc1, Hes1,* and *Pou3f3*). The differences in reactivation of developmental genes between adult and juvenile kidneys may underlie the age-dependent decline of reparative capacity of mouse and human kidneys. Future studies testing the proximal tubular heterogeneity and spatiotemporal dynamics in additional renal injury models in young and aged animals may offer further insights into molecular mechanisms governing proximal tubular cell plasticity and identify therapeutic targets.

It has been largely believed that ferroptotic stress reduces functional renal epithelial cells by intercellular propagation of ferroptotic cell death (synchronized cell death) and induces so-called necroinflammation (*Linkermann et al., 2014*; *Friedmann Angeli et al., 2014*; *Li et al., 2019*; *Strunz et al., 2020*). Consistent with this notion, we observed the accumulation of TUNEL$^+$ tubular epithelial cells in cKO kidneys. In addition to inducing ferroptosis in some tubular epithelial cells, to our surprise, our genetic knockout studies showed that excess ferroptotic stress in regenerating PT cells drives the accumulation, but not reduction, of damage-associated PT cells that augment renal inflammation. Specific gene-expression signatures indicate that damage-associated PT cells are not merely severely injured cells on the pathway to cell death but a unique functional cell state. The cells are enriched for expression of renal developmental genes such as *Sox4* and *Sox9*. SOX9 is a previously described transcription factor essential for renal repair (*Kang et al., 2016*; *Kumar et al., 2015*), and SOX4 regulates epithelial mesenchymal transition in different disease contexts (*Tiwari et al., 2013*). Moreover, damage-associated PT cells are actively involved in renal inflammation by interacting with myeloid cells through producing cytokines and chemokines. Thus, ferroptotic stress not only promotes the alteration of cell state but makes it irreversible, leading to the pathologic accumulation of cells that actively produce inflammatory and fibrogenic signals.

In summary, our study broadens the roles of ferroptotic stress from one that is restricted to the induction of regulated cell death (ferroptosis) to include the promotion and accumulation of a pathologic cell state, processes that underlie maladaptive repair. Understanding the molecular mechanisms by which ferroptotic stress controls these processes in vivo would open a new avenue for currently available and prospective anti-ferroptotic reagents to enhance tissue repair/regeneration in

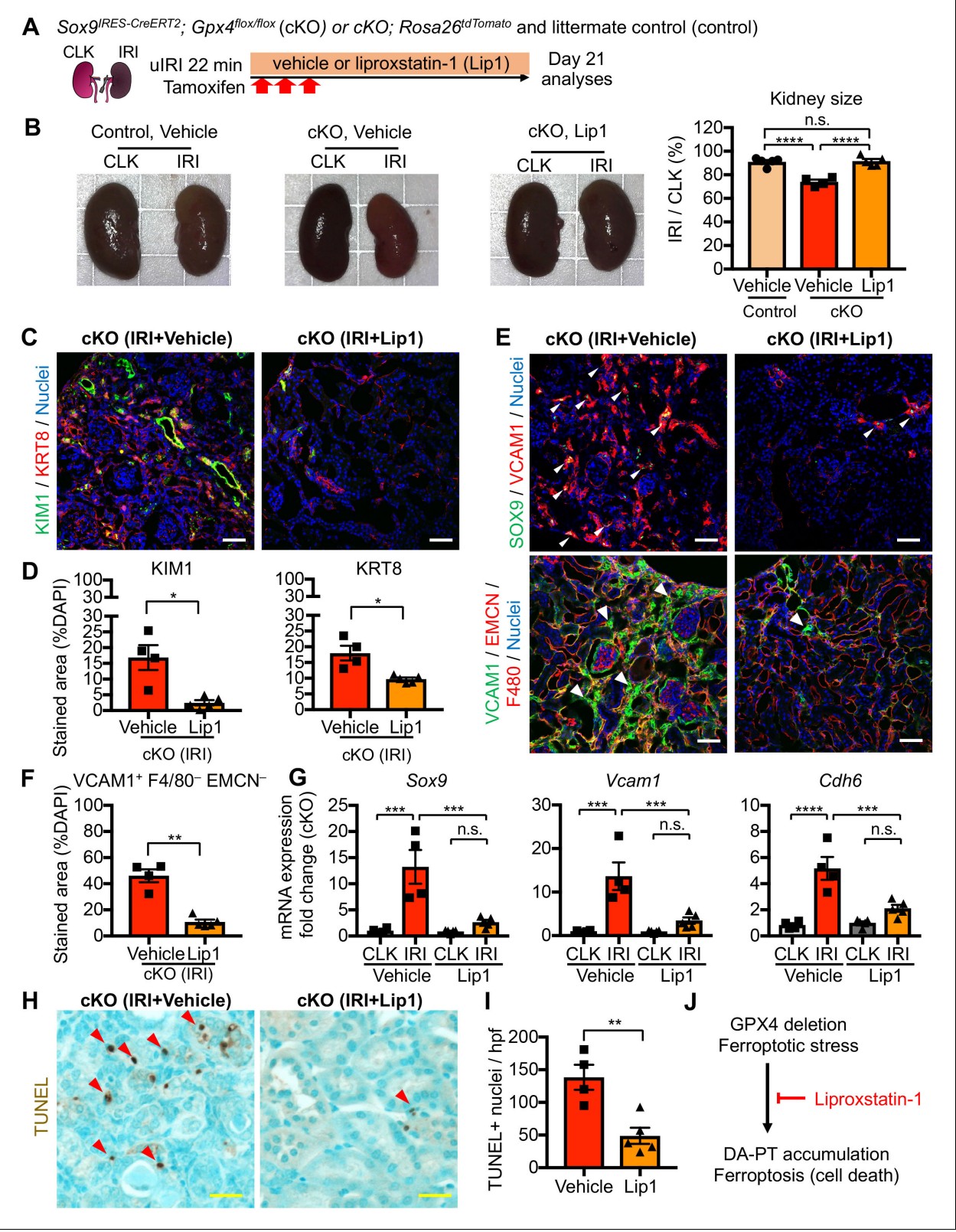

**Figure 9.** Pharmacological inhibition of ferroptotic stress blunts the accumulation of damage-associated PT cells and cell death. (**A**) Schematic representation of experimental workflow. All mice (cKO and control littermates) were subjected to the same ischemic stress (ischemic time, 22 min, unilateral IRI) and tamoxifen treatment. The same volume of vehicle was administered to the control groups (control vehicle and cKO vehicle). Kidneys were harvested on day 21 post-IRI. (**B**) Liproxstatin-1 prevents renal atrophy. Relative size of post-IRI kidneys compared to contralateral kidneys (CLK)

*Figure 9 continued on next page*

*Figure 9 continued*

was quantified. Control, littermate control. N = 4–5. (**C and D**) Immunostaining for KIM1 and KRT8. IRI kidneys from cKO are shown. Quantification of immunostained area over the DAPI⁺ area is shown in (**D**). N = 4–5. (**E and F**) Immunostaining for SOX9 and VCAM1. Quantification of VCAM1⁺EMCN⁻F4/80⁻ area over the DAPI⁺ area is shown in (**F**). Arrowheads indicate damage-associated PT cells. (**G**) Real-time PCR analyses of indicated gene expression. Whole kidney lysates were used. N = 4–5. (**H**) and (**I**) TUNEL staining for evaluating cell death. Quantification of TUNEL-positive nuclei is shown in (**I**). N = 4–5. Red arrowheads indicate TUNEL+ tubular epithelial cells. Scale bars, 50 µm in (**C**) and (**E**); and 20 µm in (**H**). *$p < 0.05$; **$p < 0.01$; ***$p < 0.001$; ****$p < 0.0001$, unpaired t-test for (**D**), (**F**). and (**I**); One-way ANOVA with post hoc multiple comparisons test for (**G**). (**J**) Liproxstatin-1 improves renal repair after IRI.

The online version of this article includes the following figure supplement(s) for figure 9:

**Figure supplement 1.** Liproxstatin-1 potently reduced ferroptotic stress in the absence of GPX4.

**Figure supplement 2.** Liproxstatin-1 potently mitigated ferroptotic stress-induced pathologic changes in the absence of GPX4.

multiple organs. Our studies provide a scientific foundation for future mechanistic and translational studies to enhance renal repair and regeneration by modulating anti-ferroptotic stress pathways to prevent AKI to CKD transition in patients.

# Materials and methods

## Key resources table

| Reagent type (species) or resource | Designation | Source or reference | Identifiers | Additional information |
|---|---|---|---|---|
| Genetic reagent (*M. musculus*) | C57BL/6J | The Jackson laboratory | RRID:IMSR_JAX:020940 | |
| Genetic reagent (*M. musculus*) | *Sox9^IRESCreERT2* | The Jackson laboratory | RRID:MGI:4947114 | |
| Genetic reagent (*M. musculus*) | *Rosa26^tdTomato* | The Jackson laboratory | RRID:IMSR_JAX:007914 | |
| Genetic reagent (*M. musculus*) | *Gpx4^flox* | The Jackson laboratory | RRID:IMSR_JAX: 027964 | |
| Antibody | Anti-SOX9 (Rabbit monoclonal) | Abcam (ab196450) | RRID:AB_2665383 | Clone EPR14335 IF: 1:200 |
| Antibody | Anti-SOX9 (Rabbit monoclonal) | Abcam (ab185966) | RRID:AB_2728660 | Clone EPR14335-78 IF: 1:200 |
| Antibody | Anti-KIM1 (goat polyclonal) | R and D systems (AF1817) | RRID:AB_2116446 | IF: 1:400 |
| Antibody | Anti-NGAL (rat monoclonal) | Abcam (ab70287) | RRID:AB_2136473 | IF: 1:400 |
| Antibody | Anti-GPX4 (Rabbit monoclonal) | Abcam (ab125066) | RRID:AB_10973901 | Clone EPNCIR144 IF: 1:200 |
| Antibody | Anti-F4/80 (Rat monoclonal) | Bio-Rad (MCA497) | RRID:AB_2098196 | Clone C1:A3-1 IF: 1:200 |
| Antibody | Anti-Endomucin (Rat monoclonal) | Abcam (ab106100) | RRID:AB_10859306 | Clone V.7C7.1 IF: 1:200 |
| Antibody | Anti-KRT8 (Rat monoclonal) | DSHB (TROMA-I) | RRID:AB_531826 | IF: 1:200 |
| Antibody | Anti-αSMA (mouse monoclonal) | Sigma (C6198) | RRID:AB_476856 | Clone 1A4 IF: 1:200 |
| Reagent, commercial | LTL | Vector laboratories (B-1325 and FL-1321) | RRID:AB_2336558 | IF: 1:200 |
| Antibody | Anti-MDA (rabbit polyclonal) | Abcam (Ab6463) | RRID:AB_305484 | IF: 1:200 |
| Antibody | Anti-ACSL4 (rabbit monocolonal) | Abcam (Ab204380) (Ab155282) | RRID:AB_2714020 | Clone: EPR8640 IF: 1:200 |

*Continued on next page*

*Continued*

| Reagent type (species) or resource | Designation | Source or reference | Identifiers | Additional information |
|---|---|---|---|---|
| Antibody | Anti-VCAM1 (rabbit monocolonal) | CST 39036S 39301S | RRID:AB_2799146 | Clone: D8U5V IF: 1:100 |
| Commercial assay, kit | RNAScope probe-Mm-Cdh6 | Advance Cell Diagnosis (Cat. 519541) | | |
| Commercial assay, kit | RNAscope Intro Pack 2.5 HD Reagent Kit Brown Mm | Advance Cell Diagnosis (Cat. 322371) | | |
| software, algorithm | ImageJ | NIH, Bethesda, MD (Version 1.52P) | RRID:SCR_003070 | https://imagej.nih.gov/ij/ |
| Software, algorithm | GraphPad Prism | | RRID:SCR_002798 | https://www.graphpad.com/scientific-software/prism/ |
| Software, algorithm | Seurat | | RRID:SCR_016341 | *Stuart et al., 2019* https://satijalab.org/seurat/get_started.html |
| Software, algorithm | Monocle 3 | | RRID:SCR_018685 | *Cao et al., 2019* https://cole-trapnell-lab.github.io/monocle3/ |
| Software, algorithm | Velocyto.R | | | *La Manno et al., 2018* https://github.com/velocyto-team/velocyto.R |
| Software, algorithm | NicheNet | | | *Browaeys et al., 2020* https://github.com/saeyslab/nichenetr/blob/master/vignettes/seurat_wrapper.md |
| Software, algorithm | RStudio | | RRID:SCR_000432 | http://www.rstudio.com/ |
| Commercial reagnet | Liberase | Roche (291963) | | 0.3 mg/ml |
| Commercial reagnet | Hyaluronidase | Sigma (H4272) | | 10 µg/mL |
| Commercial reagnet | Trypsin | Corning (45000–664) | | 0.25% |
| Chemical compound, drug | Tamoxifen | Sigma (T5648) | | 100 mg/kg |
| Chemical compound, drug | Liproxstatin-1 | Selleckchem (S7699) | | 10 mg/kg |
| Chemical compound, drug | Aristolochic acid | Sigma (A9451) | | 6 mg/kg |
| Commercial assay, kit | TUNEL staining | Abcam (Ab206386) | | |

## Animals

All animal experiments were approved by the Institutional Animal Care and Use Committee at Duke University and performed according to the IACUC-approved protocol (A051-18-02 and A014-21-01) and adhered to the NIH Guide for the Care and Use of Laboratory Animals. The following mouse lines were used for our study; *Sox9$^{IRES-CreERT2,}$* (*Furuyama et al., 2011*), *Rosa26$^{tdTomato}$* (Jackson lab, stock #007914), (*Madisen et al., 2010*), *Gpx4$^{flox}$* (Jackson lab, stock# 027964), (*Yoo et al., 2012*), and C57BL/6J (Jackson lab, stock #000664). Mice were backcrossed into a C57BL/6J background at least three times and maintained in our specific-pathogen-free facility. Timed deletion of the *Gpx4* gene and fate-mapping was achieved using *Sox9$^{IRES-CreERT2}$* knock-in mouse line with three doses of intraperitoneal injections of tamoxifen (100 mg/kg body weight, Sigma, St. Louis MO) on alternate days. The first dose of tamoxifen was administered immediately before the surgical intervention. All tested animals were included in data analyses, and outliers were not excluded. To avoid confounding

effects of age and strain background, littermate controls were used for all phenotype analyses of genetically modified mouse lines. Animals were allocated randomly into the experimental groups and analyses. The operators were blinded to mouse genotypes when inducing surgical injury models. To determine experimental sample sizes to observe significant differences reproducibly, data from our previous studies were used to estimate the required numbers. The number of biological replicates is represented by N in each figure legend. Experiments were performed on at least three biological replicates.

## Mouse models of renal injury and repair

Adult male mice aged between 8 and 16 weeks were used for all the models described below. The mice were euthanized, and kidneys were harvested for analyses. For the unilateral IRI (uIRI) model, ischemia was induced by the retroperitoneal approach on the left kidney for 20 min (mild IRI), 22 min (mild IRI in cKO studies), or 30 min (severe IRI) by an atraumatic vascular clip (Roboz, RS-5435, Gaithersburg, MD), as previously reported (*Nezu et al., 2017*; *Fu et al., 2018*). Mice were anesthetized with isoflurane and provided preemptive analgesics (buprenorphine SR). The body temperature of mice was monitored and maintained on a heat-controlled surgical pad. For the bilateral IRI (bIRI) model, ischemia was induced by the retroperitoneal approach on both kidneys for 22 min. The mice were received intraperitoneal injections of 500 µl of normal saline at the end of surgery. For the unilateral ureteral obstruction (UUO) model, the left ureter was tied at the level of the lower pole of the kidney, and the kidneys were harvested on day 10. For the aristolochic acid nephropathy (AAN) model, we used acute and chronic models, as we previously described (*Ren et al., 2020*). For the acute AAN model, three doses of 6 mg/kg body weight aristolochic acid (Sigma, A9451) in phosphate-buffered saline (PBS) were administered daily intraperitoneally to the male mice. For the chronic AAN model, six doses of 6 mg/kg body weight aristolochic acid in phosphate-buffered saline (PBS) were administered on alternate days over 2 weeks intraperitoneally to the male mice. The same volume of PBS was injected to control animals (*Ren et al., 2020*; *Dickman et al., 2011*). Contralateral kidneys (CLK), sham-treated kidneys, and vehicle-injected kidneys were used as controls depending on the models used. The numbers and dates of treatment are indicated in the individual figure legends and experimental schemes. Operators were blinded to mouse genotypes when inducing surgical injury models.

## Pharmacological inhibition of ferroptosis

Mice were randomly assigned to vehicle (1% dimethyl sulfoxide in phosphate-buffered saline) and liproxstatin-1 (10 mg/kg, Selleckchem, S7699, *Friedmann Angeli et al., 2014*) groups. Liproxstatin-1 and vehicle were administered daily by intraperitoneal injections starting from 1 hr before renal ischemia. All the mice were subjected to the same ischemic stress (22 min ischemic time, unilateral IRI model) and tamoxifen treatment. The mice were euthanatized, and kidneys were harvested on day 21 after IRI.

## Droplet-based scRNA-seq

Mice were transcardially perfused with ice-cold PBS, and the kidneys were harvested. The kidneys were dissociated with liberase TM (0.3 mg/mL, Roche, Basel, Switzerland, #291963), hyaluronidase (10 µg/mL, Sigma, H4272), DNaseI (20 µg/mL) at 37°C for 40 min, followed by incubation with 0.25% trypsin EDTA at 37°C for 30 min. Trypsin was inactivated using 10% fetal bovine serum in PBS. Cells were then resuspended in PBS supplemented with 0.01% bovine serum albumin. Our protocol yielded high cell viability (>95%) and very few doublets, enabling us to avoid the use of flow cytometry-based cell sorting. After filtration through a 40 µm strainer, cells at a concentration of 100 cells/µl were run through microfluidic channels along with mRNA capture beads and droplet-generating oil, as previously described (*Kobayashi et al., 2020*; *Macosko et al., 2015*). cDNA libraries were generated and sequenced using HiSeq X Ten with 150 bp paired-end sequencing. Each condition contains the cells from three mice to minimize potential biological and technical variability.

## Data preprocessing, unsupervised clustering, and cell type annotation of Drop-Seq data

Analysis of the scRNA-seq of mouse kidneys was performed by processing FASTQ files using drop-SeqPipe v0.3 and mapped on the GRCm38 genome reference with annotation version 91. Unique molecular identifier (UMI) counts were then further analyzed using an R package Seurat v.3.06 for quality control, dimensionality reduction, and cell clustering (*Stuart et al., 2019*). The scRNA-seq matrices were filtered by custom cutoff (genes expressed in >3 cells and cells expressing more than 500 and less than 3000 detected genes were included) to remove potential empty droplets and doublets. Relationships between the number of UMI/cell and genes/cell were comparable across the condition (*Figure 1—figure supplement 3A*). After quality control filtration and normalization using SCTransform (*Hafemeister and Satija, 2019*), UMI count matrices from post-IRI kidneys and homeostatic kidneys were integrated using Seurat's integration and label transfer method, which corrects potential batch effects (*Stuart et al., 2019*). The integrated dataset was used for all the analyses. To remove an additional confounding source of variation, the mitochondrial mapping percentage was regressed out. The number of principal components (PC) for downstream analyses were determined using elbow plot to identify knee point, and we included the first 25 PCs for the downstream analyses. A graph-based clustering approach in Seurat was used to cluster the cells in our integrated dataset. The resolution was set at 1.0 for the mouse integrated dataset. Cluster-defining markers for each cluster were obtained using the Seurat FindAllMarkers command (genes at least expressed in 25% of cells within the cluster, log fold change> 0.25) with the Wilcoxon Rank Sum test (*Supplementary file 1*). Based on the marker genes and manual curation of the gene expression pattern of canonical marker genes in UMAP plots (*Figure 1—figure supplement 4*), we assigned a cell identity to each cluster. Ambiguous clusters were shown as unknown. We manually combined 3 clusters of differentiated proximal tubular cells (PT, S1/S2 and PT, S2/S3; *Figure 1—figure supplement 4*) into one cluster (PT) to generate a more coarse-grained cell-type annotation and data visualization. We also combined three clusters of endothelial cells (Endo-1, Endo-2, and Endo-3; *Figure 1—figure supplement 4*) into one cluster (Endo) for data visualization.

## Data preprocessing, unsupervised clustering, and cell type annotation of mouse neonatal kidneys

The RDS files for mouse neonatal kidneys (postnatal day 1) were obtained from Gene Expression Omnibus (GEO accession number: GSE94333, GSM2473317), (*Adam et al., 2017*). Data were analyzed as in our mouse kidney dataset using Seurat and SCTransform (*Stuart et al., 2019*; *Hafemeister and Satija, 2019*). We included the first 17 PCs for the downstream analyses of mouse neonatal kidneys. A graph-based clustering approach in Seurat was used to cluster the cells. The resolution was set at 0.8. Based on the marker genes and manual curation of the gene expression pattern of canonical marker genes in UMAP plots (*Figure 1—figure supplement 7*), we assigned a cell identity to each cluster. The anchor genes for assigning cell identity were obtained from previous single-cell transcriptome analyses of the developing mouse kidneys (*Adam et al., 2017*; *Combes et al., 2019b*).

## Data preprocessing, unsupervised clustering, and cell type annotation of human kidneys

The RDS files for human kidneys were obtained from Gene Expression Omnibus (GEO accession number: GSE131882 and GSE145927), (*Malone et al., 2020*; *Wilson et al., 2019*). Normal human kidney data was originated from two macroscopically normal nephrectomy samples without renal mass (GSE131882; GSM3823939 and GSM3823941), (*Wilson et al., 2019*). Human AKI kidney data was originated from two biopsy-samples of transplant kidneys with evidence of AKI and acute tubular injury but no evidence of rejection (GSE145927; GSM4339775 and GSM4339778), (*Malone et al., 2020*). Data were integrated and analyzed as in the mouse kidney analyses using Seurat's integration method and SCTransform (*Stuart et al., 2019*; *Hafemeister and Satija, 2019*). We included the first 25 PCs for the downstream analyses of human normal and AKI kidneys. A graph-based clustering approach in Seurat was used to cluster the cells. The resolution was set at 0.5 for normal human kidneys and 1.0 for the human AKI kidneys. Based on the marker genes and manual curation of the gene expression pattern of canonical marker genes in UMAP plots (*Figure 5—figure supplement 2*

and *Figure 6—figure supplement 1*), we assigned a cell identity to each cluster. The anchor genes for assigning cell identity were obtained from previous single-cell transcriptome analyses of the human kidneys (*Malone et al., 2020*; *Wilson et al., 2019*; *Stewart et al., 2019*).

## Differential gene expression analyses and Gene ontology (GO) enrichment analyses

To predict the cellular functions based on enriched gene signature, we performed gene-ontology enrichment analyses. Differentially expressed genes obtained using FindMarkers command in Seurat were used for identifying signaling pathways and gene ontology through Enricher (*Supplementary file 2* and *3*; *Figure 1—figure supplement 5B*), (*Kuleshov et al., 2016*). To visualize the overrepresented signaling pathways, scaled data in the integrated Seurat object were extracted. Then, mean values of the scaled score of gene members in each GO class were calculated and shown in UMAP (*Kobayashi et al., 2020*). The gene member lists of signaling pathways were obtained from AmiGO 2 (*AmiGO Hub et al., 2009*). Log$_2$ fold changes and *P*-values of each gene extracted using FindMarkers command in Seurat with Wilcoxon rank sum test were shown in a volcano plot using an R package EnhancedVolcano v1.4.0 (*Blighe et al., 2021*; https://github.com/kevinblighe/EnhancedVolcano), (*Figure 1—figure supplement 5B*). Top 100 genes in mature and early PT cell clusters were obtained using the 'FindMarkers' command in Seurat. These genes were visualized on the UMAP plots using the scaled score as in GO class visualization.

## RNA velocity analyses

To infer future states of individual cells, we performed RNA velocity analyses (*La Manno et al., 2018*) using single-time point dataset of post-IRI kidney on day 7. The aligned BAM files were used as input for Velocyto to obtain the counts of unspliced and spliced reads in loom format. Cell barcodes for the clusters of interests (PT and DA-PT) were extracted and utilized for velocyto run command in velocyto.py v0.17.15, as well as for generating RNA velocity plots using velocyto.R v0.6 in combination with an R package SeuratWrappers v0.2.0 (*Stuart et al., 2019*; https://github.com/satijalab/seurat-wrappers). Twenty-five nearest neighbors in slope calculation smoothing were used for RunVelocity command.

## Pseudotime trajectory analyses

To infer the dynamic cellular process during injury and repair, we performed single-cell trajectory analyses. We first extracted the clusters of interests (PT and DA-PT) from our integrated Seurat object of mouse kidneys and utilized for Monocle 3 (version 0.2.3.0) analyses with default parameters to identify a pseudotime trajectory with SeuratWrappers v0.2.0 (*Cao et al., 2019*; *Trapnell et al., 2014*). We set the starting states in two different approaches. We used the UMAP space area occupied by cells from the earliest time point of IRI kidneys (6 hr post-IRI, *Figure 1F*) and the area occupied by the cells with high expression of genes that are highly expressed in differentiated PT cells, such as *Slc34a1* (*Figure 1—figure supplement 8B*) as the starting state, respectively. Both approaches resulted in similar trajectory inference. For the human AKI dataset, we extracted the clusters of interests (PT and DA-PT-like) from our integrated Seurat object and applied the Monocle 3 algorithm with default parameters. We used the UMAP space area occupied by the cells with high expression of homeostatic genes (*ALDOB*), (*Figure 6F*).

## Intercellular communication analyses using NicheNet

To predict the intercellular communication process between damage-associated PT (DA-PT) cells and myeloid cells (monocytes and macrophages), we performed NicheNet analyses based on the analytical pipeline (*Browaeys et al., 2020*; https://github.com/saeyslab/nichenetr/blob/master/vignettes/seurat_wrapper.md) using an R package nichenetr (version 1.0.0) with default parameters (*Browaeys et al., 2020*). Based on high enrichment of chemokines and cytokines in DA-PT cells and the observed positive association between the numbers of macrophages and DA-PT cells in severely injured kidneys, we surmised that they have a close molecular interaction. We used NicheNet to predict the ligand-receptor pairs that are most likely to explain the target gene expression in renal myeloid cells after IRI. We defined DA-PT cells as the 'sender/niche' cell population and myeloid cells as the 'receiver/target' cell population in our integrated Seurat object for these analyses. We defined

the differentially expressed genes in monocytes or macrophages in IRI-kidneys compared to homeostatic kidneys as the gene sets of interest that were affected by predicted ligand-receptor interactions.

## Tissue collection and histology

Kidneys were prepared as described previously (*Nezu et al., 2017*; *Ide et al., 2020*). For cryosections (7 µm), the tissues were fixed with 4% paraformaldehyde in PBS at 4°C for 4 hr and then processed through a sucrose gradient. Kidneys were embedded in OCT compound for sectioning. For paraffin sections (5 µm), the tissues were fixed with 10% neutral buffered formalin overnight at 4°C and processed at Substrate Services Core and Research Support at Duke. Sections were blocked (animal-free blocker with 0.5% triton x-100) for 30 min and incubated with the primary antibodies overnight at 4°C. Primary antibodies used were as follows: SOX9 (Abcam, Cambridge, UK, ab196450 or ab185966, 1:200), KIM1 (R and D Systems, Minneapolis, MN, AF1817, 1:400), NGAL (Abcam, ab70287, 1:400), F4/80 (Bio-rad, Hercules, CA, MCA497G, 1:200), α-SMA (Sigma, C6198, 1:200), LTL (Vector, Burlingame, CA, B-1325 or FL-1321, 1:200), KRT8 (DSHB, TROMA-I, 1:200), MDA (Abcam, ab6463, 1:200), ACSL4 (Abcam, ab204380 or ab155282, 1:200), EMN (Abcam, 106100, 1:200), VCAM1 (CST, 39036S or 33901S, 1:100), and GPX4 (Abcam, ab125066, 1:200). Alexa Fluor-labeled secondary antibodies were used appropriately for immunofluorescence. ImmPRES HRP reagent kit was used for immunohistochemistry (Vector, MP-7401). Nuclei were stained with DAPI (1:400, Sigma). Heat-induced antigen retrieval was performed using pH 6.0 sodium citrate solution (eBioscience). Experiments for RNAScope in situ hybridization (Advanced Cell Diagnostics, ACD, Newark, CA) was performed as recommended by the manufacturer. Mm-Cdh6 (ACD, 519541) was used. Images were captured using Axio imager and 780 confocal microscopes (Zeiss, Oberkochen, Germany). Paraffin-sections were stained with hematoxylin and eosin (H and E). The kidney injury score was calculated as we previously reported (*Ren et al., 2020*). TUNEL staining was performed following the manufacturer's instruction (Abcam, ab206386). To ensure the TUNEL signal's specificity, we used sections treated with DNase I as a positive control and a section treated without terminal deoxynucleotidyl transferase as a negative control, as recommended by the manufacturer. Sections were counterstained with methyl green. More than three randomly selected areas from at least three kidneys were imaged and quantified using ImageJ (*Ide et al., 2020*). The stitched large area was used for quantification to alleviate the selection bias in the acquisition of images. All representative images were from more than three kidneys tested.

## RNA extraction and real-time quantitative PCR

Total RNA was extracted from kidneys using the TRIzol reagent (Invitrogen, 15596026). Three µg of total RNA was then reverse transcribed with Maxima H minus cDNA synthesis master mix (Invitrogen, M1662). Equivalent amounts of diluted cDNA from each sample were analyzed with Real-time PCR with the primers listed below using the Powerup SYBR Green reagent (Invitrogen, A25776) on a QuantStudio three real-time PCR systems (Thermo). 18S rRNA expression was used to normalize samples using the $\Delta\Delta$CT-method.

## Statistical analysis

Statistical analyses were conducted using GraphPad Prism software. Two-tailed unpaired Student's t-test was used for two groups, and one-way analysis of variance (ANOVA) followed by Sidak multiple comparison test was used for more than two groups. All results are represented as means ± SE. A p value less than 0.05 was considered statistically significant.

Additional protocols are available in the supplementary method.

Primers used for quantitative PCR.

*Sox9*: Fw-GAGCCGGATCTGAAGAGGGA, Rv-GCTTGACGTGTGGCTTGTTC
*Vcam1*: Fw-TCTTACCTGTGCGCTGTGAC, Rv-ACTGGATCTTCAGGGAATGAGT
*Cdh6*: Fw-CCAATATTCACCAAGGACGTTTA, Rv-CGTGACTTGGACCACAAATG
*Acsm2*: Fw-CCAAGATGGCAGAACACTCC, Rv-TCAGAAGTACTCAGGCCTGTCC
*Icam1*: Fw-GCTACCATCACCGTGTATTCG, Rv-AGGTCCTTGCCTACTTGCTG
*Pdgfb*: Fw-CGAGGGAGGAGGAGCCTA, Rv-GTCTTGCACTCGGCGATTA
*Apoe*: Fw-TTGGTCACATTGCTGACAGG, Rv-AGCGCAGGTAATCCCAGAA
*Havcr1*: Fw-AAACCAGAGATTCCCACACG, Rv-GTCGTGGGTCTTCCTGTAGC

*Lcn2*: Fw-CAAGCAATACTTCAAAATTACCCTGTA, Rv-GCAAAGCGGGTGAAACGTT
*Acta2*: Fw-CCCACCCAGAGTGGAGAA, Rv-ACATAGCTGGAGCAGCGTCT
*Slc34a1*: Fw-CTCATTCGGATTTGGTGTCA, Rv-GGCCTCTACCCTGGACATAGA
*Krt8*: Fw-CTGAGCTTGGCAACATGC, Rv-ACGCTTGTTGATCTCATCCTC
*18S rRNA*: Fw-CGGCTACCACATCCAAGGAA, Rv-GCTGGAATTACCGCGGCT

Genotyping primers.

*Cre*, Fw: GTGCAAGTTGAATAACCGGAAATGG,
Cre, Rv: AGAGTCATCCTTAGCGCCGTAAATCAAT
*Gpx4 $^{flox}$*, wt: CTGCAACAGCTCCGAGTTC
*Gpx4 $^{flox}$*, common: CGGTGCCAAAGAAAGAAAGT
*Gpx4 $^{flox}$*, mut: CCAGTAAGCAGTGGGTTCTC
*Rosa26$^{tdTomato}$*, Fw: CTGTTCCTGTACGGCATGG
*Rosa26$^{tdTomato}$*, Rv-GGCATTAAAGCAGCGTATCC
*Rosa26$^{wt}$*, Fw: AAGGGAGCTGCAGTGGAGTA
*Rosa26$^{wt}$*, Rv: CCGAAAATCTGTGGGAAGTC.

## Acknowledgements

We thank Drs. Brigid Hogan and Myles Wolf for critical advice and helpful suggestions on the manuscript. We also thank Dr. Helene F Kirshner (Duke Center for Genomic and Computational Biology) for her bioinformatical support on our single-cell transcriptome dataset. The monoclonal antibody against keratin 8 (TROMA-1, developed by Drs. P Brulet and R Kemeler) was obtained from the Developmental Studies Hybridoma Bank, created by the NICHD of the NIH and maintained at the Department of Biology, The University of Iowa. We thank Drs. Tetsuhiro Yokonishi (Duke University), Leslie Gewin and Kensei Taguchi (Vanderbilt University), and members of the Crowley lab for their technical advice. This study was supported by grants from the National Institute of Diabetes and Digestive and Kidney Diseases (R01 DK123097), a pilot award from the Northwestern University George M O'Brien Kidney Research Core Center (P30 DK114857), the American Society of Nephrology Carl W Gottschalk Career Developmental Grant, and Duke Nephrology Start-up Fund to TS. SI, YK, and KI are supported in part by fellowship grants from the American Heart Association, Japan Society for the Promotion of Science, and the Astellas Foundation for Research on Metabolic Disorders, respectively. Imaging was performed at the Duke Light Microscopy Core Facility supported by the shared instrumentation grant (1S10RR027528-01).

## Additional information

### Funding

| Funder | Grant reference number | Author |
| --- | --- | --- |
| National Institute of Diabetes and Digestive and Kidney Diseases | R01 DK123097 | Tomokazu Souma |
| National Institute of Diabetes and Digestive and Kidney Diseases | P30 DK114857 | Tomokazu Souma |
| American Society of Nephrology | | Tomokazu Souma |
| American Heart Association | 20POST35210465 | Shintaro Ide |
| Astellas Foundation for Research on Metabolic Disorders | | Kana Ide |
| Japan Society for the Promotion of Science | | Yoshihiko Kobayashi |

The funders had no role in study design, data collection and interpretation, or the decision to submit the work for publication.

## Author contributions
Shintaro Ide, Formal analysis, Investigation, Writing - original draft, Writing - review and editing; Yoshihiko Kobayashi, Formal analysis, Investigation, Methodology, Writing - review and editing; Kana Ide, Formal analysis, Investigation, Project administration, Writing - review and editing; Sarah A Strausser, Investigation, Project administration, Writing - review and editing; Koki Abe, Formal analysis, Investigation; Savannah Herbek, Investigation, Writing - review and editing; Lori L O'Brien, Formal analysis, Writing - review and editing; Steven D Crowley, Methodology, Writing - review and editing; Laura Barisoni, Resources, Writing - review and editing; Aleksandra Tata, Resources, Methodology, Writing - review and editing; Purushothama Rao Tata, Resources, Supervision, Methodology, Writing - review and editing; Tomokazu Souma, Conceptualization, Formal analysis, Supervision, Funding acquisition, Investigation, Writing - original draft, Project administration, Writing - review and editing

## Author ORCIDs
Shintaro Ide (iD) https://orcid.org/0000-0002-9301-211X
Kana Ide (iD) http://orcid.org/0000-0002-2845-8481
Lori L O'Brien (iD) http://orcid.org/0000-0002-0741-181X
Steven D Crowley (iD) http://orcid.org/0000-0002-1838-0561
Purushothama Rao Tata (iD) http://orcid.org/0000-0003-4837-0337
Tomokazu Souma (iD) https://orcid.org/0000-0002-3285-8613

## Ethics
Animal experimentation: All animal experiments were approved by the Institutional Animal Care and Use Committee at Duke University and performed according to the IACUC-approved protocols (A051-18-02 and A014-21-01) and adhered to the NIH Guide for the Care and Use of Laboratory.

## Decision letter and Author response
Decision letter https://doi.org/10.7554/eLife.68603.sa1
Author response https://doi.org/10.7554/eLife.68603.sa2

# Additional files

## Supplementary files
- Supplementary file 1. Cluster-enriched genes in *Figure 1*.
- Supplementary file 2. Differentially expressed genes in PT cells.
- Supplementary file 3. Gene ontology analyses of PT cells.
- Transparent reporting form

## Data availability
Sequencing data have been deposited in GEO under accession codes GSE161201.

The following dataset was generated:

| Author(s) | Year | Dataset title | Dataset URL | Database and Identifier |
|---|---|---|---|---|
| Ide S, Kobayashi Y, Ide K, Strausser SA, Herbek S, O'Brien LL, Crowley SD, Barisoni L, Tata A, Tata PR, Soum T | 2020 | Ferroptotic stress promotes the accumulation of pro-inflammatory proximal tubular cells in maladaptive renal repair | https://www.ncbi.nlm.nih.gov/geo/query/acc.cgi?acc=GSE161201 | NCBI Gene Expression Omnibus, GSE161201 |

The following previously published datasets were used:

| Author(s) | Year | Dataset title | Dataset URL | Database and Identifier |
|---|---|---|---|---|
| Adam M, Potter SS | 2017 | The use of cold active proteases can dramatically reduce single cell RNA-seq gene expression artifacts | https://www.ncbi.nlm.nih.gov/geo/query/acc.cgi?acc=GSE94333 | NCBI Gene Expression Omnibus, GSE94333 |
| Wilson PC, Humphreys BD | 2019 | The Single Cell Transcriptomic Landscape of Early Human Diabetic Nephropathy | https://www.ncbi.nlm.nih.gov/geo/query/acc.cgi?acc=GSE131882 | NCBI Gene Expression Omnibus, GSE131882 |
| Malone AF | 2020 | Single Cell Transcriptional Analysis of Donor and Recipient Immune Cell Chimerism in the Rejecting Kidney Transplant | https://www.ncbi.nlm.nih.gov/geo/query/acc.cgi?acc=GSE145927 | NCBI Gene Expression Omnibus, GSE145927 |

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
