## [Decision Letter]

**Acceptance summary:**

In this study, Ide et al., present a comprehensive analysis of single cell transcriptomic changes in the kidney in response to mild and recoverable injury compared to severe and persistent injury after renal ischemia reperfusion in an effort to identify cellular pathways that promote maladaptive repair. They find that cellular pathways (Gpx4-glutathione) that prevent ferroptosis, a major pathway known to drive cell death in renal ischemia-reperfusion injury, are involved. This is further corroborated in an elegant genetic mouse model with mild ischemic stress-induced ablation of GPX4 and by pharmacologic inhibition of ferroptosis. These studies will be of significant interest both to those studying acute kidney injury and others interested in ischemic injury in other organ systems.

**Decision letter after peer review:**

Thank you for submitting your article "Ferroptotic stress promotes the accumulation of pro-inflammatory proximal tubular cells in maladaptive renal repair" for consideration by *eLife*. Your article has been reviewed by 3 peer reviewers, and the evaluation has been overseen by a Reviewing Editor and Mone Zaidi as the Senior Editor. The following individual involved in review of your submission has agreed to reveal their identity: Marcus Conrad (Reviewer #2).

Essential revisions:

1. Necessary experiments: Please test whether ferroptosis inhibitors administered in vivo blunt some of the dynamic changes/plasticity of proximal tubular cells or impair the overall damage in the genetic GPX4 null model (data presented in Figures3, 5, 7, 8). These studies were deemed necessary to corroborate the contribution of ferroptosis in the proposed model.

2. Please address the variability between the staining patterns for *SOX9*immunostaining and the *Sox9*-TdTomato reporter. For example, in Figure 2D/E, the number of *SOX9*+ cells (over DAPI+ cells) is only ~8% in the 30-minute IRI kidney at day 21 while in Figure 3B/C, the number of *Sox9*;TdTomato+ cells (over DAPI+ cells) is 60% in the cortex and 40% in the medulla in the 30-minute IRI kidney at day 21. Even in the 20-minute IRI kidney, the *SOX9*+ cells are close to 0% at day 21 (Figure 2E) but the *Sox9*;TdTomato+ cells are close to 20% in Figure 3C. We also ask that you discuss the limitations of using *SOX9* as a primary marker for maladaptive proximal tubular repair.

3. Please explain the variability of VCAM1 staining across figures. For example, in Figure S9B VCAM1 staining is almost completely negative in the contralateral control kidney while there is a fair amount of VCAM1 positive staining in Figure 2C and Figure 3D of the contralateral control kidney. Please also provide an explanation for VCAM1 expression in a subset of normal human kidney proximal tubule cells (Figure S11B).

4. Please discuss the limitations of using pseudo time and RNA velocity in RNA-Seq analysis to identify novel transitional cell states given that the results are imputed by mathematical modeling.

*Reviewer #1 (Recommendations for the authors):*

– A major concern with these studies is the reliance on *SOX9*+ cells to represent DA-PT cells. Figures S5D and 5C demonstrate that less than 40% of the DA-PT cells express *SOX9*. *SOX9* is not specific to DA-PT cells either, as they are seen in both PT cells and DCT1 cells as well. Additionally, the scale of the dot plots should be 0-100, and not 0-40. Also, Figures 5 D and E do not differentiate between DCT1 and PTs and this should be clarified.

– When defining the initial DA-PT cell population, the authors should include data showing there is no upregulation of genes associated with the cellular death pathways.

– Did the authors observe the DA-PT phenotype after "mild" 20 minute uIRI? This is relevant since the authors show that *SOX9*+ cells appear after 20 minute uIRI.

– Figure 3E would suggest that only 25% of the *SOX9*+ cells were also VCAM1+. If only 40% of DA-PT cells were *SOX9*+, that would suggest that only 10% of DA-PT cells were *SOX9*+VCAM+, again questioning the validity of using *SOX9* for identifying DA-PT cells.

– How do the authors reconcile the link between ferroptosis and severe AKI when accumulation of MDA and ACSL4 seems to occur in only a minority of *SOX9*+ cells and is also seen in *SOX9*- cells?

– Figure 5 does not include comparison of sequencing results for the mild-injury. Are there differences in the gene expression of MDA and ACSL4 in the 20 minutes vs 30 minutes uIRI?

– Human data in Figure 6E is not particularly convincing that the "DA-PT like" cells are expressing *SOX9*, VCAM or CDH6. Expression looks very low and potentially higher in normal PT cells. Violin (with individual dots) or dot blots should be used to show the quantification of these data.

– The justification for the 22 minutes of ischemia for the studies with the cKO mice in Figures 7 and 8, should be provided.

*Reviewer #2 (Recommendations for the authors):*

Suggestions to authors:

– Title: "Ferroptotic stress" sounds rather awkward as the I/R stress is the trigger of ferroptosis in kidney tubular cells.

– Page 8: Would in vivo active ferroptosis inhibitors blunt some of the dynamic changes/plasticity of proximal tubular cells (Figures 3, 5)? Along the same line, would they also impair the overall damage in the genetic GPX4 null model? These studies would be extremely helpful to corroborate the contribution of "ferroptosis" in the proposed model.

– Did the author consider secondary forms of cell death including apoptosis and necroptosis in their model as a consequence of a highly proinflammatory milieu?

*Reviewer #3 (Recommendations for the authors):*

As one of the main strengths of this manuscript is the validation of the scRNA-Seq data analysis findings, it is important that the validations are clear. Some of the immunostaining are not consistent with expected findings (Lotus lectin staining) or are variable (VCAM1 and *SOX9*), described below.

1. Figure S6B Image is not consistent with LTL staining which should be apical staining of the brush border. The staining in the image appears cytoplasmic or reflects kidney autofluorescence.

2. *SOX9* staining appears to be variable and the abundance of *SOX9*+ cells differ between *SOX9*immunostaining and the *Sox9*-TdTm reporter. In Figure 5D/F, S9B *SOX9*immunostaining in the control kidney with zero or rare positive cells while *Sox9*-tdTomato have quite a few scattered throughout the kidney given the larger field presented at a lower magnification, as shown in Figure 3B-control kidney. Depending on the field chosen, the *Sox9*-TdTm+ cells could be as high as 5-6 per higher power field and as low as 0-2 (which is what Figure 3D presents). Is the overall pattern and frequency of *SOX9*+ immunostaining at a lower power field also consistent with the *Sox9*-TdTomato reporter image in Figure 3B?

Figure 3B control kidney image also does not support the sentence on page 12 line 11 stating that *Sox9*-CreERT2 activity is "not induced" in non-injured kidneys, as there are clearly *Sox9*-TdTm+ cells in Figure 3B in the control kidney, unless the *Sox9*-CreERT2 is leaky and some of the lineage traced cells are false positives, in which case interpretation of the experiments using the *Sox9*-CreERT2 may require some consideration of this possibility.

3. There is variability of the VCAM1 staining throughout the manuscript. Figure S9B contralateral control kidney is VCAM1 staining is almost completely negative, while there is a fair amount of VCAM1 positive staining in Figure 2C and Figure 3D of the contralateral control kidney, albeit the VCAM1+ cells appear interstitial. It is described in the text that VCAM1 is "expressed weakly" in macrophages and endothelial cells after injury, although some of these images in the manuscript suggest that VCAM1 is relatively strongly expressed at baseline, and further induced after injury in interstitial and/or endothelial cells. Of note, it is odd that VCAM1 expression is present in a subset of normal human kidney proximal tubule cells, specifically presented on the Dot plot in Figure S11B. Is there an explanation for this finding?

---

## [Author Response]

Essential revisions:1. Necessary experiments: Please test whether ferroptosis inhibitors administered in vivo blunt some of the dynamic changes/plasticity of proximal tubular cells or impair the overall damage in the genetic GPX4 null model (data presented in Figures3, 5, 7, 8). These studies were deemed necessary to corroborate the contribution of ferroptosis in the proposed model.

Following the reviewer’s advice, we have performed a pharmacological inhibitor study using liproxstatin1 (a potent inhibitor of ferroptosis that scavenges lipid peroxides) in our genetic Gpx4 conditional deletion model (Sox9^IRES-CreERT2^; Gpx4 ^flox/flox^). The new data summarized below strongly support our original notion that ferroptotic stress promotes the accumulation of pathologic proximal tubular cells in addition to ferroptotic cell death, thus preventing successful repair. Targeting ferroptotic stress in acute kidney injury holds the promise of preventing maladaptive repair and improving long-term renal outcomes.

1) Liproxstatin-1 potently prevented renal atrophy after IRI in the absence of *Gpx4* (Figure 9B).

2) In conditional knockout (cKO) mice after IRI, liproxstatin-1 mitigated overall damage compared to the vehicle control (Figure 9C and 9D). Liproxstatin-1-treated cKO kidneys show reduced expression of tubular injury markers (KIM-1 and KRT8) compared to vehicle-treated cKO kidneys after IRI. We observed that *Sox9*-lineage cells (tdTomato-positive) are positive for KIM1 in the vehicle cKO group, but largely negative in liproxstatin-1-treated cKO kidneys (Figure 9—figure supplement 2B). Moreover, the Sox9lineage cells express a high level of LTL-binding (a differentiated PT cell marker) in the liproxstatin-1treated cKO group, but not in the vehicle-treated cKO group (Figure 9—figure supplement 2C).

Please note that liproxstatin-1 and vehicle-treated animals underwent the same procedures; unilateral ischemia-reperfusion (ischemic time 22 min), tamoxifen treatment, and daily intraperitoneal injections of liproxstatin-1 or vehicle.

1) Liproxstatin-1 effectively reduced the accumulation of *Sox9*+Vcam1+ proximal tubular cells in postIRI cKO kidneys (Figure 9E and 9F). Liproxstatin-1 also reduced the expression of damage-associated proximal tubular cell markers (*Sox9*, Vcam1, and Cdh6) compared to the vehicle group in post-IRI kidneys (Figure 9G).

2) Liproxstatin-1 reduced tubular cell death (TUNEL staining, Figure 9H and 9I) in cKO post-IRI kidneys.

3) Liproxstatin-1 did not affect the efficacy of genetic targeting in cKO kidneys (Figure 9—figure supplement 1). As a control experiment, we employed lineage-tracing and GPX4 immunostaining to detect the genetic targeting efficacy of our mouse models treated with liproxstatin-1. We observed a similar level of *Gpx4* deletion both in liproxstatin1-treated and vehicle-treated groups after IRI (Figure 9—figure supplement 1, B and C). Moreover, liproxstatin-1 effectively reduced expression of ACSL4, a ferroptotic stress marker, in ischemia-reperfusion injured cKO kidneys (Figure 9—figure supplement 1D).

2. Please address the variability between the staining patterns for SOX9 immunostaining and the Sox9-TdTomato reporter. For example, in Figure 2D/E, the number of SOX9+ cells (over DAPI+ cells) is only ~8% in the 30-minute IRI kidney at day 21 while in Figure 3B/C, the number of Sox9;TdTomato+ cells (over DAPI+ cells) is 60% in the cortex and 40% in the medulla in the 30-minute IRI kidney at day 21. Even in the 20-minute IRI kidney, the SOX9+ cells are close to 0% at day 21 (Figure 2E) but the Sox9;TdTomato+ cells are close to 20% in Figure 3C. We also ask that you discuss the limitations of using SOX9 as a primary marker for maladaptive proximal tubular repair.

We thank the reviewers for this comment and apologize that our data presentation was not entirely clear. We employed a combination of genetic fate-mapping and immunostaining to dissect dynamic phenotypic alteration of proximal tubular cells. The *Sox9^IRES-CreERT2^; Rosa26^tdTomato^* mouse line was used to permanently tag *Sox9*-lineage cells with tdTomato expression and thereby visualize the history of *Sox9* activation (Figure 3B and C). TdTomato expression accumulates during the course of injury and repair, but does not necessarily indicate ongoing *Sox9* expression at the time when the kidney tissue is analyzed. To detect this, we used immunostaining of *SOX9* (Figure 2 D and E). The “tdTomato positive SOX9-immunofluorescence (IF)-positive” cells thus represent cells with ongoing SOX9 activity, whereas “tdTomato-positive, SOX9IF negative” cells are those with a history of transient SOX9 expression that currently lack SOX9 activity. Combining these tools, we interpreted our results as follows: After mild ischemia-reperfusion (20 minischemia), *SOX9* is transiently induced, but the expression is diminished as cells return to their original state on day 21. However, severe ischemia induces persistent and ongoing SOX9 activation in some of the *Sox9*-lineage cells during the course of maladaptive repair.

We recognize the limitation of using *SOX9* alone as a primary marker for damage-associated PT cell state. In our case, we used a combination of markers to detect these pathological proximal tubular cells (ex. *SOX9*+VCAM1+ cells). This point is emphasized in our revised manuscript (page 7, line 25, page 8, line 10, and page 9, line 3).

3. Please explain the variability of VCAM1 staining across figures. For example, in Figure S9B VCAM1 staining is almost completely negative in the contralateral control kidney while there is a fair amount of VCAM1 positive staining in Figure 2C and Figure 3D of the contralateral control kidney. Please also provide an explanation for VCAM1 expression in a subset of normal human kidney proximal tubule cells (Figure S11B).

We thank the reviewers for carefully reviewing our manuscript. We observed VCAM1 expression in endothelial cells and macrophages in normal kidneys, but not in the tubular epithelial cells. VCAM1 is induced in proximal tubular cells after injury and serves as a marker of damage-associated PT cell state. We believe this inconsistency in the previous Figure S9 was due to the confocal imaging setting, which was slightly different when we captured these images. We have obtained new images, analyzed the data, and replaced the images for this figure (See Figure 2—figure supplement 1). The expression pattern of the new images is consistent with Figure 2C and 3D.

VCAM1 expressing human proximal tubular cells in the previous Figure S11B (current Figure 5—figure supplement 2) have been identified as a scattered cell population in human kidneys (PMID: 23124355). A recent report by Muto et al., investigated this unique subset of proximal tubular cells (PMID: 33850129). They used single-nucleus ATAC and RNA sequencing and identified that NFkB signaling is activated in this subset and promoting VCAM1 expression. The physiological and pathological roles of this subset of proximal tubular cells are still unclear and require further investigation.

4. Please discuss the limitations of using pseudo time and RNA velocity in RNA-Seq analysis to identify novel transitional cell states given that the results are imputed by mathematical modeling.

We agree with the reviewers. The results are imputed by mathematical modeling and require extensive validation, such as genetic fate-mapping combined with immunostaining. Following the reviewer’s advice, we included this discussion in our revised manuscript (Page 8, line 1 and 5).

Reviewer #1 (Recommendations for the authors):– A major concern with these studies is the reliance on SOX9+ cells to represent DA-PT cells. Figures S5D and 5C demonstrate that less than 40% of the DA-PT cells express SOX9. SOX9 is not specific to DA-PT cells either, as they are seen in both PT cells and DCT1 cells as well. Additionally, the scale of the dot plots should be 0-100, and not 0-40. Also, Figures 5 D and E do not differentiate between DCT1 and PTs and this should be clarified.

Markers for DA-PT cell state: We thank the reviewer for raising this important point regarding the use of multiple markers to define DA-PT cells. We recognize the limitation of using *SOX9* alone as a marker of damage-associated PT cell state and performed a series of validation studies using multiple markers, including SOX9, VCAM1, and CDH6.

Dot plot Scale for Figure 5C: Single-cell and single-nucleus RNA sequencing has been implemented successfully in multiple disease processes and has identified heterogeneity of gene expressions. Many investigators use the default setting of Seurat for dot plot data representations with adjusted scale ranges (ex. Aviv Regev and Maria Lehtinen’s group, Figure 7, Cell 2021: 184, 3056-3074; PMID: 33932339). We would like to keep the current data representation as it is visually recognizable. However, we increased the font size of scales to highlight the heterogeneity.

Differentiation between DCT1 and PT: We agree with this reviewer on this point. Due to the limited antibody combinations, it is technically challenging to differentiate DCT1 and PT cells in Figure 5 at this time. Instead, we used our genetic *Gpx4* knockout mouse studies to show the functional importance of ferroptotic stress in regulating proximal tubular cell fate (i.e. accumulation of Vcam1+ cells. Vcam1 is not induced in DCT1 cells). Our data collectively support our notion that ferroptotic stress in DA-PT cells promotes the accumulation of this pathologic state, cell death, and drives the maladaptive renal repair.

– When defining the initial DA-PT cell population, the authors should include data showing there is no upregulation of genes associated with the cellular death pathways.

Following the reviewer’s suggestion, we analyzed the expression pattern of genes associated with necroptosis (*Ripk3* and *Mlkl*) and pyroptosis (*caspase-1* and *Gsdmd*). DA-PT cells do not have high expression levels of these genes (Author response image 1).

**Author response image 1. sa2fig1:** UMAP plots showing genes associated with necroptosis and pyroptosis. Blue arrowheads indicate DA-PT cells. These genes are not highly expressed in the DA-PT cell population.

– Did the authors observe the DA-PT phenotype after "mild" 20 minute uIRI? This is relevant since the authors show that SOX9+ cells appear after 20 minute uIRI.

We observed transient appearance of SOX9+VCAM1+ cells in the kidneys that underwent 20 min ischemia as shown in Figure 2—figure supplement 1B. However, these markers were negative in proximal tubular cells on day 21 (Figure 2—figure supplement 1B).

– Figure 3E would suggest that only 25% of the SOX9+ cells were also VCAM1+. If only 40% of DA-PT cells were SOX9+, that would suggest that only 10% of DA-PT cells were SOX9+VCAM+, again questioning the validity of using SOX9 for identifying DA-PT cells.

We apologize that our data presentation and description were not clear as they should be. We also agree with the reviewer that we need to use multiple markers to define the DA-PT cell state.

As described in the essential revision section (#2), the data of Figure 3E show the percentage of DA-PT cells over the *Sox9*-lineage cells (but not current SOX9-expressing cells on day 21 post-IRI). The tdTomatopositive cells are the cells with a “history” of *Sox9* expression. The number of tdTomato+ cells accumulates over the experimental time course. The expression of tdTomato does not indicate the current SOX9 expression nor DA-PT state at the time when the kidneys were harvested. Conversely, VCAM1immunopositivity indicates these cells are in a DA-PT state at the time of harvest.

– How do the authors reconcile the link between ferroptosis and severe AKI when accumulation of MDA and ACSL4 seems to occur in only a minority of SOX9+ cells and is also seen in SOX9- cells?

We would like to thank this reviewer again for raising this important question. *Sox9*-lineage cells have been identified as a critical cellular population for successful renal repair (PMID: 26279573 and 26776520). Therefore, we hypothesized that the damage to these cells has a high impact on failed renal repair. Indeed, genetic induction of ferroptotic stress selectively in these cells drove the maladaptive repair process (Figure 7: increased tubular injury, inflammation, and fibrosis). Moreover, our new data using a ferroptosis inhibitor demonstrate that we can prevent these pathologic changes induced by genetic deletion of *Gpx4* in ischemia-reperfusion-injured kidneys (Figure 9). Collectively, our data demonstrate the critical role of ferroptotic stress in SOX9+ cells in maladaptive repair process.

To answer the potential roles of ferroptotic stress in SOX9-negative tubular epithelial cells, we are currently generating a new mouse line and would like to address this intriguing question in future studies.

– Figure 5 does not include comparison of sequencing results for the mild-injury. Are there differences in the gene expression of MDA and ACSL4 in the 20 minutes vs 30 minutes uIRI?

We did not observe a statistically significant difference in gene expression for *Acsl4* between 20 min vs. 30 min on day 1 post-IRI (data not shown). However, we observed the difference at a protein level (Author response image 2), confirming the previous observation that ACSL4 protein expression serves as a quantitative marker of ferroptosis and ferroptotic stress (Cell Mol Life Sci 2017; PMID 28551825).

Malondialdehyde (MDA) is a reactive aldehyde and marker for lipid peroxidation, commonly quantified using immunostaining. Following the reviewers’ comments, we analyzed MDA expression by immunostaining. As expected, severe ischemia (30 min) showed higher MDA expression compared to 20 min mild ischemia.

**Author response image 2. sa2fig2:** Immunostaining and quantification for malondialdehyde (MDA) and ACSL4. The post-IRI kidneys harvested on 6-hrs and 1-day after ischemia were used to detect MDA and ACSL4, respectively. *P<0.05 and ***P<0.001 unpaired t-test. Scale bars, 50µm. Note tha 30-min ischemia induces more ferroptotic stress (MDA and ACSL4) in damaged kidneys compares to 20-min ischemia.

– Human data in Figure 6E is not particularly convincing that the "DA-PT like" cells are expressing SOX9, VCAM or CDH6. Expression looks very low and potentially higher in normal PT cells. Violin (with individual dots) or dot blots should be used to show the quantification of these data.

We thank the reviewers for this comment to improve our data visualization. Following the reviewer’s suggestion, we used dot plots to represent our data (Figure 6E). Our analyses showed that at least three cellular states exist in human proximal tubular cells after acute kidney injury (state1, 2, and 3). We observed a decreasing trend of differentiated PT cell markers from state 1 to state 3 and an increasing trend of damage-induced genes from state 1 to state 2 and state 3. State 3 has high damage-induced genes and low homeostatic differentiation markers, indicating that cells in state 3 are closely related to the mouse DA-PT cell state. We agree with this reviewer that a relatively small fraction of cells in state 3 (DA-PT-like) expresses *SOX9*, *VCAM1* and *CDH6*. This result could be species-specific or reflect the time of biopsy in these two AKI patients. The potential of species-specific transcriptional responses to AKI of various etiologies needs further investigation with a larger cohort of patients in the future.

– The justification for the 22 minutes of ischemia for the studies with the cKO mice in Figures 7 and 8, should be provided.

We found that an ischemic time of 22 min is the most reliable protocol to induce genetic deletion of *Gpx4* in *Sox9^IRESCREERT2^; Gpx4^flox/flox^*. In this protocol, we found that the kidneys with control genotype (*Gpx4^flox/flox^*) manifest a successful repair as in the 20 min ischemia protocol. Therefore, we used 22 min instead of 20 min for this mouse line. To minimize the effect of genetic background, we only used littermate controls to compare the phenotypic differences.

Reviewer #2 (Recommendations for the authors):Suggestions to authors:– Title: "Ferroptotic stress" sounds rather awkward as the I/R stress is the trigger of ferroptosis in kidney tubular cells.

We thank the reviewers for this comment. As summarized in the essential revision section (#1), we now show that a ferroptosis inhibitor, liproxstatin-1, effectively mitigates the accumulation of damage associated proximal tubular cells and cell death (ferroptosis) after IRI in the absence of *Gpx4* in *Sox9*-lineage cells. Our results emphasize the critical role of ferroptotic stress in dynamic phenotypic alterations of proximal tubular cells among the various stressors evoked by IRI. Therefore, we respectfully request to keep our title in its current form.

– Page 8: Would in vivo active ferroptosis inhibitors blunt some of the dynamic changes/plasticity of proximal tubular cells (Figures 3, 5)? Along the same line, would they also impair the overall damage in the genetic GPX4 null model? These studies would be extremely helpful to corroborate the contribution of "ferroptosis" in the proposed model.

We thank the reviewer for these insightful and clinically important comments. As summarized in the essential revision section (#1), we used liproxstatin-1 to treat our conditional knockout mice (*Sox9^IRESCreERT2^; Gpx4 ^flox/flox^*) that underwent unilateral IRI. While the vehicle and IRI-treated cKO kidneys showed maladaptive repair phenotype, liproxstatin-1 potently mitigated tubular injury and reduced the accumulation of pathologic proximal tubular cells (DA-PT cells, SOX9+VCAM1+) and cell death (ferroptosis). These data strongly support our initial notion that ferroptotic stress triggers both cell death and pathologic cellular changes in proximal tubular cells. We consider glutathione peroxidase 4 to be a central guardian for the proximal tubular cell fate.

– Did the author consider secondary forms of cell death including apoptosis and necroptosis in their model as a consequence of a highly proinflammatory milieu?

We thank this reviewer for raising this interesting question. It is highly likely that other types of regulated cell death are also induced in the hostile microenvironment of failed repair model. We would like to interrogate how multiple forms of regulated cell death may interact and contribute to failed renal repair in future studies.

Reviewer #3 (Recommendations for the authors):As one of the main strengths of this manuscript is the validation of the scRNA-Seq data analysis findings, it is important that the validations are clear. Some of the immunostaining are not consistent with expected findings (Lotus lectin staining) or are variable (VCAM1 and SOX9), described below.

We thank the reviewer for their positive comments on our extensive validation and carefully reviewing our manuscript.

1. Figure S6B Image is not consistent with LTL staining which should be apical staining of the brush border. The staining in the image appears cytoplasmic or reflects kidney autofluorescence.

We thank the reviewer for carefully reviewing our manuscript. We have retaken the images and re-analyzed the data accordingly (previous Figure S6B is renumbered to Figure 1—figure supplement 6B).

2. SOX9 staining appears to be variable and the abundance of SOX9+ cells differ between SOX9 immunostaining and the Sox9-TdTm reporter. In Figure 5D/F, S9B SOX9 immunostaining in the control kidney with zero or rare positive cells while Sox9-tdTomato have quite a few scattered throughout the kidney given the larger field presented at a lower magnification, as shown in Figure 3B-control kidney. Depending on the field chosen, the Sox9-TdTm+ cells could be as high as 5-6 per higher power field and as low as 0-2 (which is what Figure 3D presents). Is the overall pattern and frequency of SOX9+ immunostaining at a lower power field also consistent with the Sox9-TdTomato reporter image in Figure 3B?

We apologize that our data representation was not clear as it should be. As summarized in the essential revision section (#2), SOX9 immunostaining and *Sox9*-tdTomato-expression represent different cellular states. SOX9 immunostaining detects the ongoing SOX9 activity, but tdTomato expression identifies the history of *Sox9* activation. We observed rare SOX9-immunopositive cells in normal and contralateral kidneys, and we identified more *Sox9*-lineage cells in these conditions as the genetic fate-mapping system detects all the cells that expressed SOX9 during the experimental time course as a sum. Therefore, the tdTomato+ area (history of *Sox9* activity) is larger than the immunostained area (ongoing *Sox9* activity).

Additionally, we have added quantification of *Sox9*-lineage area of contralateral kidneys in new Figure 3C.

Regarding the overall pattern of SOX9+ cells, we observed a consistent expression pattern between immunostaining and lineage-mapping data. SOX9 expression is mostly confined to distal convoluted tubule both in lineage-reporter and immunostaining for control kidneys (Author response image 3).

**Author response image 3. sa2fig3:** Distal convoluted tubules express *SOX9* protein in uninjured kidneys. *Sox9*-lineage tagged uninjured kidneys were used to localize *Sox9*-lineage cells (tdTomato). Sodium chloride co-transporter (NCC, Slc12a3) is a marker for distal convoluted tubular cells. Insets: Individual fluorescent channels of dotted area are shown. Scale bar: 20 µm. Arrows: double-positive cells.

Figure 3B control kidney image also does not support the sentence on page 12 line 11 stating that Sox9-CreERT2 activity is "not induced" in non-injured kidneys, as there are clearly Sox9-TdTm+ cells in Figure 3B in the control kidney, unless the Sox9-CreERT2 is leaky and some of the lineage traced cells are false positives, in which case interpretation of the experiments using the Sox9-CreERT2 may require some consideration of this possibility.

We agree with this reviewer that our description was not accurate. In control kidneys, *Sox9* is only expressed in a small subset of distal convoluted tubular cells (Author response image 3) but not in proximal tubular cells. To more precisely describe this point, we rephrased “*Sox9*-CreERT2 activity is not induced in non-injured kidneys” to “*Sox9*-CreERT2 activity is not induced in non-injured proximal tubular cells” (Page 12, line 23).

3. There is variability of the VCAM1 staining throughout the manuscript. Figure S9B contralateral control kidney is VCAM1 staining is almost completely negative, while there is a fair amount of VCAM1 positive staining in Figure 2C and Figure 3D of the contralateral control kidney, albeit the VCAM1+ cells appear interstitial. It is described in the text that VCAM1 is "expressed weakly" in macrophages and endothelial cells after injury, although some of these images in the manuscript suggest that VCAM1 is relatively strongly expressed at baseline, and further induced after injury in interstitial and/or endothelial cells. Of note, it is odd that VCAM1 expression is present in a subset of normal human kidney proximal tubule cells, specifically presented on the Dot plot in Figure S11B. Is there an explanation for this finding?

We thank the reviewer for carefully reviewing our manuscript. As described in the essential revision section (#3), we have corrected the previous Figure S9B (current Figure 2—figure supplement 1). Regarding the previous Figure S11B (current Figure 5—figure supplement 2), The expression of VCAM1 in a subset of proximal tubular cells has been reported (J Pathol 2013, PMID: 23124355) and interrogated in detail recently (Nat Comm 2021, PMID; 33850129).